

# Tracking slow-moving landslides with PlanetScope data: new perspectives on the satellite's perspective

Ariane Mueting[1] and Bodo Bookhagen[1]

[1]Institute of Geosciences, University of Potsdam, Karl-Liebknecht-Str. 24-25, Potsdam-Golm 14476, Germany

**Correspondence:** Ariane Mueting (mueting@uni-potsdam.de)

**Abstract.** PlanetScope data with daily temporal and 3-m spatial resolution hold an unprecedented potential to quantify and monitor surface displacements from space. Slow-moving landslides, however, are complex and dynamic targets that alter their topography over time. This leads to orthorectification errors, resulting in inaccurate displacement estimates when images acquired from varying satellite perspectives are correlated. These errors become particularly concerning when the magnitude of

orthorectification error exceeds the signal from surface displacement which is the case for many slow-moving landslides with annual velocities of 1 - 10 m/yr. This study provides a comprehensive assessment of orthorectification errors in PlanetScope imagery and presents effective mitigation strategies for both unrectified L1B and orthorectified L3B data. By implementing these strategies, we achieve sub-pixel accuracy, enabling the estimation of realistic and temporally coherent displacement over landslide surfaces. The improved signal-to-noise ratio results in higher-quality disparity maps, allowing for a more detailed

analysis of landslide dynamics and their driving factors.

## 1 Introduction

Optical image offset tracking is a standard method for quantifying horizontal surface displacement caused by landslides (e.g., Stumpf et al., 2014, 2017; Lacroix et al., 2019; Dille et al., 2021), glaciers (e.g., Kääb et al., 2016; Gardner et al., 2018; Lei et al., 2021; Aati et al., 2022a), earthquakes (e.g., Leprince et al., 2007; Kääb et al., 2017; Feng et al., 2019; Socquet et al.,

2019; Aati et al., 2022a), and other geoscientific processes that offset land surfaces over time. The conceptual approach relies on two or more consecutive images of a common area of interest (AOI) and detects displacements between them. Optical offset measurements are complementary to line-of-sight measurements from radar interferometry: they are sensitive to both north-south (NS) and east-west (EW) components and can track meter-scale displacements in the image plane without losing coherence. Images with a high spatial resolution of < 5 m enable the detection of small offsets and extensive satellite and

airphoto archives allow analyses of ground motions to be extended to the last decades (Milliner et al., 2016; Andreuttiova et al., 2022). Recent advancements in satellite technologies, especially CubeSat, have led to an increase in the number of high-quality instruments in orbit (Mehrparvar et al., 2014). CubeSats have the ability to provide optical images at both high spatial and temporal resolutions, reducing the limitations inherent to optical data: higher spatial resolution allows for the detection of finer-scale movements, while daily coverage increases the likelihood of obtaining cloud-free imagery. Remotely sensed data

hold great potential for studying Earth surface processes in remote and challenging terrain, where installing permanent Global



Navigational Satellite System (GNSS) stations or cameras is impractical due to access difficulties. In these regions, image cross-correlation is a cost-effective technique to provide insights into past and recent surface displacement.

In this study, we explore the potential of optical PlanetScope data to study surface displacements related to slow-moving landslides. PlanetScope currently represents the largest commercial Earth observation satellite constellation in orbit and provides daily optical acquisitions at 3 m spatial resolution (Planet, 2022b). PlanetScope data offer new opportunities for understanding landslide dynamics and land deformation processes (Mazzanti et al., 2020; Hermle et al., 2021; Dille et al., 2021; Muhammad et al., 2022). However, the use of high-resolution optical data in displacement analysis poses various technical challenges, particularly when precise sub-pixel measurements and accuracies are required. One of the main limitations in utilizing optical data for displacement analysis is the inherent relative geolocation accuracy. The relative geolocation accuracy refers to the positional alignment of scenes capturing the same area and can be affected by specific geometric distortions, sensor noise, and calibration errors of the acquiring instruments. While the relative geolocation accuracy of PlanetScope images is typically below 2 pixels (6 m) (Planet, 2022a), landslides are prone to orthorectification errors that locally degrade the co-registration between scenes, resulting in substantial biases in derived disparity estimates in these areas. Orthorectification errors are not exclusive to the PlanetScope constellation and have been documented for other satellite systems, such as Sentinel-2 cross-track pairs (Kääb et al., 2016; Chudley et al., 2022). PlanetScope scenes, however, are acquired by different sensors at different view angles and positions which makes their offset-tracking results highly susceptible to bias related to outdated Digital Elevation Model (DEM) heights.

This study focuses on assessing the impact of orthorectification errors on displacement estimates derived from PlanetScope scenes acquired from different perspectives and proposes strategies to mitigate them. With the satellite perspective, we describe the satellite's view that is jointly determined by the satellite's look direction, view angle, and motion direction. By carefully selecting correlation pairs acquired from common perspectives, performing manual orthorectification using updated DEM surfaces, and accounting for systematic ramp errors and stereoscopic effects, we are able to improve the signal-to-noise ratio and bring the estimated accuracy for displacement measurements into the sub-pixel range. This improvement in accuracy is essential for a more comprehensive assessment of the dynamics of slow-moving landslides through image cross-correlation. Slow-moving landslides typically exhibit annual velocities that are below the uncertainties introduced by orthorectification errors and other factors that compromise the relative geolocation accuracy. By addressing and minimizing these sources of error, image cross-correlation can provide valuable information on the behavior and movement of slow-moving landslides.

Slow-moving landslides are an important component of the geomorphic and natural hazard system. Due to ongoing climate changes and stochastic earthquake occurrences, they have large impacts on local communities, infrastructure, and sediment-transport regimes (e.g., Mansour et al., 2011). Of particular interest are the driving factors for landslide generation and acceleration – these are often coupled to extreme rainfall or seismic events (e.g., Keefer, 2002; Hilley et al., 2004; Lacroix et al., 2015; Handwerger et al., 2019, 2022). Insights on the mechanisms controlling slow-moving landslides can be translated to catastrophic landslides whose physical parameters can rarely be studied during failure (Lacroix et al., 2020b). In this work, we exemplify the analysis of surface displacement rates with PlanetScope imagery for two geographic locations from the western




and eastern Andes: The Siguas landslide is located in Peru near Arequipa on the Pacific Coast (Hermanns et al., 2012; Lacroix et al., 2019, 2020a; Graber et al., 2021), and the Del Medio landslide in the eastern Andes in northwestern Argentina near the town of Jujuy (Savi et al., 2016; Purinton and Bookhagen, 2018). Both landslides have continued to accumulate displacement of several meters per year following their initial failures in 2005 and 2009 and are characteristic examples of dynamic Earth

surface processes. The presented methodology, however, can be transferred to other targets and regions to improve the detection of surface-deformation patterns.

## 2 Test cases and their geographic setting

We investigate two exemplary slow-moving landslides with velocities between ~ 2 - 12 m/yr located on the western flank of the central Andes in southern Peru and in the eastern central Andes in northwestern Argentina. In order to test and validate

our approaches, we rely on previously identified and described settings. We have selected our test sites as they are (1) situated in different terrains (flat vs. steep mountains), (2) differ in annual velocities by approximately an order of magnitude, (3) are located in (semi-)arid environments which improve matching capabilities due to the absence of vegetation, and (4) have experienced substantial elevation changes since the acquisition of the SRTM DEM in the year 2000, making these sites particularly vulnerable to orthorectification errors. The results of this work are transferable to other regions and targets. Both study sites

are described in more detail below.

### 2.1 Del Medio landslide

The Del Medio landslide is situated at the southern end of the Humahuaca basin at approximately -65.56° longitude and -23.93° latitude (Figure 1 A, B). The transition from the Andean foreland (~ 1500 m) to the internally-drained Altiplano-Puna plateau with elevations in excess of ~ 4000 m is characterized by steep terrain with high erosion rates, sparse vegetation cover, and

frequent rainstorm events (e.g., Bookhagen and Strecker, 2012; Castino et al., 2017). These conditions in combination with pre-existing fault structures and active tectonic deformation (Strecker et al., 2007; Figueroa et al., 2021) are ideal for slope movement processes including debris flows and landslides (e.g., Savi et al., 2016; Purinton and Bookhagen, 2018; Mueting et al., 2021). The Del Medio catchment maintains an aggrading debris flow fan that has been active throughout at least the last 400 years (Savi et al., 2016) and debris flows sourced from the Del Medio catchment have repeatedly damaged local

infrastructure and blocked the Río Grande (e.g., Marcato et al., 2009), the main river draining the Humahuaca basin. The sediment supply for debris-flow activity in the Del Medio catchment is largely generated by landslides and rockfalls that have left visible scars throughout the catchment (Savi et al., 2016). Most visible is the landslide scarp and deposits of a rockfall that occurred in 2009 in the upper part of the catchment at the contact of two lithologic units, suggesting a structural control of the landslide. Results of this study indicate that the Del Medio landslide has continued to creep downhill after the initial rapid

rockfall incident in 2009.



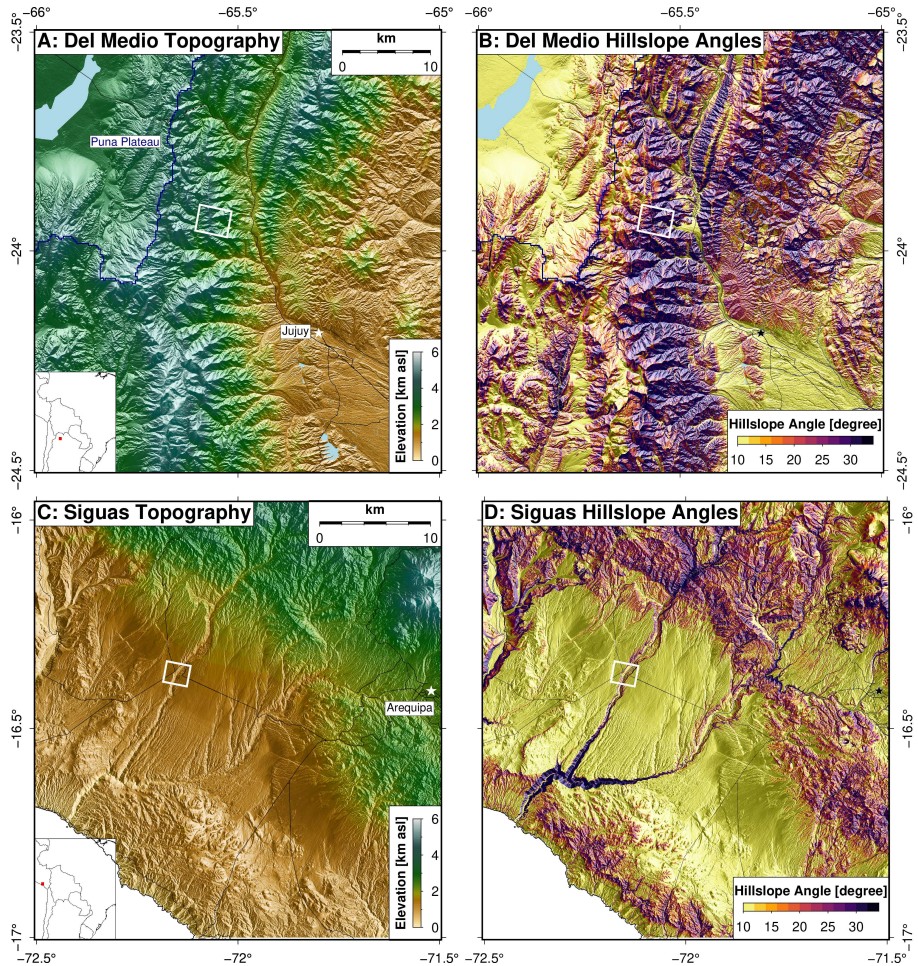

**Figure 1.** Copernicus 30 m topography of the Del Medio (A, B) and Siguas (C, D) landslides. Major cities are indicated by white stars and the extent of the study area is shown by a white box. Black lines show the major road network and the blue line in A and B outlines the eastern border of the Central Andean Plateau. Hillslope angles derived from the Copernicus DEM show the steep terrain in the eastern Andes (B) and the low-sloping alluvial plain in southwestern Peru (D).

## 2.2 Siguas landslide

The Siguas landslide is located in southern Peru at approximately -72.15° longitude -16.37° latitude (Figure 1 C, D) and has been previously described by Hermanns et al. (2012); Lacroix et al. (2019, 2020a); Graber et al. (2021). It sits at the rim of the Siguas river valley which is deeply incised into a wide, low-sloping alluvial plateau. The Majes-Siguas irrigation project
has transformed the region since the 1980s bringing water to the plateau above the river valley and converting the otherwise arid surface into agricultural land. The increased infiltration has affected groundwater conditions and triggered several large landslides in the early 2000s, including the Siguas landslide (Hermanns et al., 2012; Lacroix et al., 2020a; Graber et al., 2021).





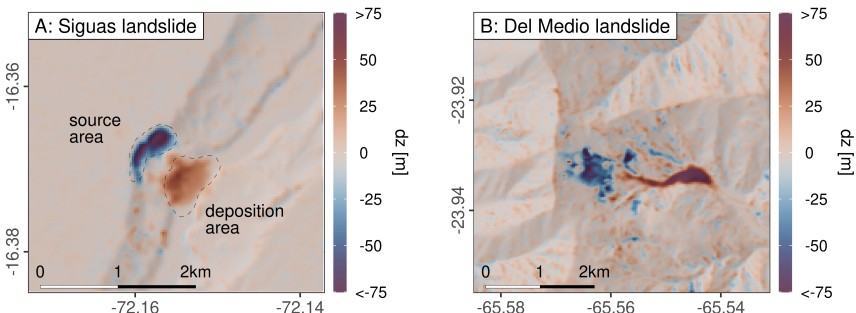

**Figure 2.** Elevation difference between the SRTM DEM (reprocessed version now called NASADEM) from 2000 (NASA JPL, 2021) and the Copernicus DEM from 2012-2015 (European Space Agency, 2021) across the Siguas landslide (A) and the Del Medio landslide (B). Both landslides have accumulated elevation changes of more than 75 m as a result of rapid rockfall events and gradual landslide motion. The DEMs were aligned using *demcoreg* (Shean et al., 2016). The source and deposition areas of the Siguas landslide were manually outlined to assess the propagation of orthorectification error to the disparity derived from PlanetScope scenes as discussed under 4.2.

Since its initiation in 2005, the Siguas landslide has continued to move downhill at temporally variable rates with peak velocities of up to 35 m/yr (Lacroix et al., 2019). The landslide movement follows a self-entrainment process including successive phases of headscarp retrogression and sliding of the main landslide body (Lacroix et al., 2019).

## 3 The PlanetScope Constellation

The PlanetScope constellation, operated by Planet (Planet Labs PBC, San Francisco, CA, USA) consists of multiple generations of 3U CubeSats, called "Doves". Approximately 130 of these doves fly in a sun-synchronous orbit at 475 - 525 km height and continuously monitor the Earth's surface (Planet, 2022b). Their unique in-sequence arrangement captures consecutive images while the Earth rotates below (Kääb et al., 2017; Planet, 2022b). PlanetScope satellites carry a telescope and a charged coupled device (CCD) array, which converts electromagnetic radiation into electronic signals (Kääb et al., 2017). Images are captured in a push-frame acquisition mode – a technique where filters are used to divide the CCD array into sub-frames, allowing only electromagnetic radiation of specific spectral wavelengths to pass. For the oldest generation of PlanetScope doves with the instrument ID PS2 (also known as Dove-C), the CCD array is divided by a visible and Near-Infrared (NIR) bandpass filter (Planet, 2022a, 2023). A Bayer pattern filter, covering the entire CCD sensor allows the separation of red, green, and blue (RGB) bands. NIR measurements are stored at the green pixels of the RGB Bayer-mask (Planet, 2022a). The final 4-band scene is obtained by merging the RGB image acquired by the top half of the image with the NIR acquisition from an adjacent frame covering the same area (Planet, 2023), and will therefore be half the size of the original CCD array. The RGB bands are thus acquired simultaneously from the same satellite position while the NIR band is captured at a different time and a slightly different satellite position (Kääb et al., 2017; Planet, 2022a). This is an important consideration for image distortion and misalignment.



For the newer PlanetScope generations, PS2.SD (Dove-R) and PSB.SD (SuperDove), the Bayer pattern filter was replaced with a butcher-block filter for improved optical quality (Aati et al., 2022a; Planet, 2023). The filter divides the image frame into four (PS2.SD) or eight (PSB.SD) homogeneous sub-frames capturing reflected electromagnetic radiation at different wavelengths.

A composite scene is then produced by aligning and merging sub-frames of a common wavelength on both sides of a given frame (Aati et al., 2022a; Planet, 2023). In contrast to PS2 scenes, the newer generation images are not acquired at a single camera position. PS2.SD and PSB.SD scenes are a composite of stripes acquired by a moving satellite that were aligned using a homography estimated from matching features across an $\sim 8\%$ overlap, assuming flat terrain (Aati et al., 2022a). While this stitching process allows producing larger and sharper images (Bayer pattern interpolation is avoided), it may also introduce

artifacts which will be further discussed in this work. The archive of PS2 data dates back to 2016; PS2.SD data are available since March 2019. Both generations have been decommissioned in April 2022, leaving PSB.SD as the only active imaging PlanetScope instrument in orbit at the time of writing (July 2023). The earliest available PSB.SD images are from March 2020 (Planet, 2022b).

PlanetScope scenes are available as rectified and unrectified assets. Level 1B (L1B) data represent the rawest form accessible

for download to the general user. The image is stored as scaled Top of Atmosphere Radiance as seen from the satellite in space (Planet, 2022b). Geometric adjustments at the sensor are limited to the correction of spacecraft-related effects through attitude telemetry and ephemeris data, and refinement using ground control points (GCPs) (Planet, 2022b). GCPs are derived by matching PlanetScope scenes to optical reference images (e.g., the National Agricultural Image Program (NAIP), the sensor onboard the Advanced Land Observing Satellite (ALOS), Landsat, and other high-resolution images) (Planet, 2022b). L1B

scenes are not projected to a cartographic projection – users are provided with Rational Polynomial Coefficients (RPCs) to enable orthorectification without revealing the intrinsic sensor parameters. RPCs are the coefficients of the Rational Functional Model (RFM) (or RPC Model) which is a replacement model for the physical sensor model (Tao and Hu, 2001). The RFM allows transforming between 2D image- and 3D object-space coordinates by modeling the transformation as a ratio of two third-order polynomials (Tao and Hu, 2001; Grodecki and Dial, 2003). The RFM has become a common tool to describe the

geometry of optical images and has been widely adopted for CubeSat constellations (Aati and Avouac, 2020). It is important to note that the RFM is only an approximation of the true transformation and may not be sufficiently accurate for tasks such as 3D extractions or precise georeferencing (Aati and Avouac, 2020). Several studies have been conducted to compensate for the inherent bias in the RFM (e.g., Grodecki and Dial, 2003; de Franchis et al., 2014; Aati and Avouac, 2020; Aati et al., 2022a), and Aati and Avouac (2020); Aati et al. (2022a) specifically address RFM refinement of PlanetaryScope imagery. The current

RPCs supplied with PlanetScope L1B data provide a positional accuracy of less than 10 m RMSE (Planet, 2022b).

For users seeking pre-processed and ready-to-use georeferenced imagery, Planet provides the Level-3B (L3B) data product. L3B scenes have undergone a correction for terrain distortions and were projected to a cartographic coordinate reference system. For orthorectification, Planet uses DEMs derived from different sensors, e.g., SRTM, Intermap, and local elevation datasets (Planet, 2022b).





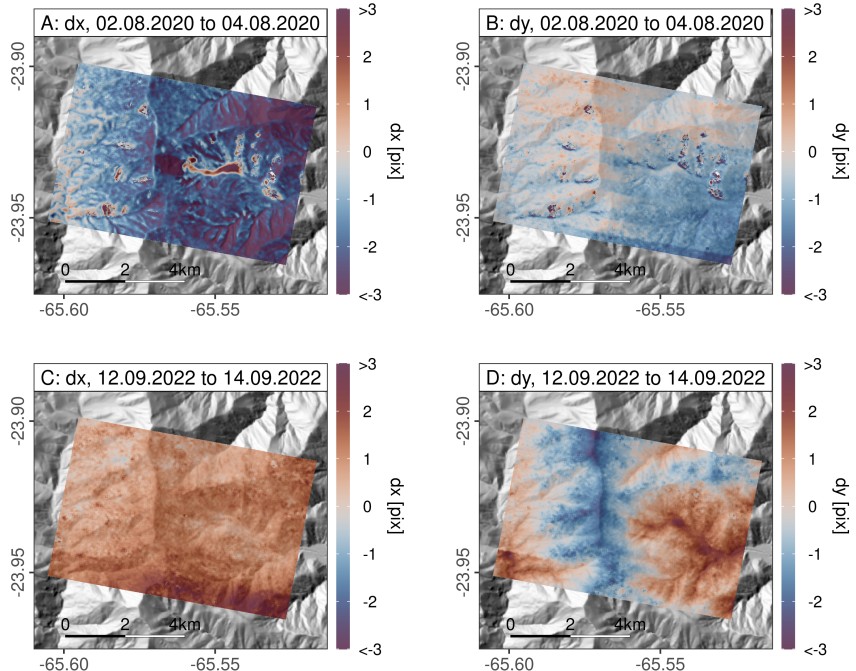

**Figure 3.** Disparity maps displaying the East-West (dx) and North-South component (dy) for two PlanetScope L3B image pairs with a temporal baseline of 2 days for the Del Medio test site. A, B: 02.08.2020 and 04.08.2020; C, D: 12.09.2022 and 14.09.2022. There are no surface displacements at this short time scale and we use the disparity maps to highlight common problems in the orthorectified L3B data, particularly in high-relief environments: (1) Outdated DEMs used for orthorectification (A); (2) stereoscopic effects that were not properly corrected (D); (3) global shifts between scenes (A, C); and (4) striping due to misalignment between sub-frames composing a full scene (B, C, D). The striping is particularly strong in the dy component and in the earliest (approx. first half of 2020) PSB.SD scenes (B). The alignment process seems to have improved for newer acquisitions. The disparity maps in C and D solely indicate misalignment of the lowermost sub-frame.

## 3.1 Relative geolocation accuracy

The geolocation accuracy of a PlanetScope scene is influenced by several factors, including the number, quality, and distribution of tiepoints identified between PlanetScope and reference imagery, as well as the roll angle of the satellite during acquisition (Planet, 2019). In the most recent quality report, the relative geolocation accuracy of PlanetScope scenes has been reported as 6.15 m at the 90th percentile of RMSEs for PSB.SD and 1.72 m for PS2 instruments (Planet, 2022a). While a $\sim 6$ m geolocation accuracy will be sufficient for many applications, inherent pixel shifts exceeding the annual displacement rate of many slow-moving landslides will severely affect the estimated displacement. Additional geometric corrections are therefore inevitable to obtain more accurate results through improved scene-to-scene alignment (Frazier and Hemingway, 2021). Apart from the quality of the GCP registration, another significant factor impacting the relative geolocation accuracy between two





optical images is lateral distortion related to DEM errors introduced during the orthorectification process. These errors may
result from the DEM not accurately representing the real-world topography (due to DEM errors or limited spatial resolution)
or from an actual change in topography if a DEM is not acquired concurrently with the optical image projected onto it, which
is rarely the case (Kääb et al., 2017). Both of these factors are particularly relevant to landslides or glaciers. Landslides are
commonly found in mountainous terrain where steep cliffs may be smoothed in a global DEM with 30 m resolution (e.g., Hirt,
2018; Purinton and Bookhagen, 2021). Additionally, mountainous regions are prone to DEM errors resulting from effects like
foreshortening or occlusions (e.g., Purinton and Bookhagen, 2017). Even more significant is the fact that landslides naturally
alter the topography over time. Although slow-moving landslides change the terrain at a slower pace than rapid events, a DEM
obtained years before the optical imagery may not reflect the current topography if sufficient displacement has accumulated.
This issue would be minimized if all images were distorted equally. But since PlanetScope imagery is acquired by different
satellites at different positions from different perspectives, variable distortions will occur. In consequence, image pixels across
the landslide are prone to orthorectification errors which further reduce the relative geolocation accuracy in this area.

Orthorectification errors are a particular problem at the Siguas and Del Medio landslides, because the initial failures in 2005
and 2009 and subsequent displacement caused significant elevation changes to the terrain (Figure 2 B) that are not recorded
in DEMs acquired before that date. When using Planet orthorectified L3B data, the user will not know which DEM was used
during the orthorectification process. According to the Planet Product Specifications (Planet, 2022b), Level 3B data are or-
thorectified using terrain models from different sources, which are periodically updated. In our investigation of PlanetScope
data over the Siguas and Del Medio sites, we assume that most scenes were orthorectified based on the SRTM DEM from 2000,
so there is a $\sim$ 20-year gap between the DEM acquisition and the images that are projected onto it. The elevation changes that
occurred during that time resulted in significant lateral distortions: disparity maps derived from L3B data show an apparent
displacement signal at the order of $\pm$ 3 pixels (9 m) across the landslide area (Figure 3 A), even in pairs with minimal temporal
baseline. Distortions of this magnitude make it nearly impossible to retrieve an accurate estimate of the true displacement that
has occurred.

Another artifact in the PlanetScope PS2.SD and PSB.SD instruments are stripes that appear in the disparity maps, particularly
in the NS direction, due to misregistration of the sub-frames that make up the composite scene (Aati et al., 2022a) (Figure 3
B-D). These stripes are absent from PS2 images, as the entire scene corresponds to a single frame. Specifically, scenes acquired
by the earliest PSB.SD instruments are affected (Figure 3 B). In more recent PSB.SD imagery, the sub-frame alignment has
significantly improved and the transition appears much smoother (Figure 3 D). For some image pairs subtle striping is still
visible, causing artificial offsets at the order of $\sim$ 1 pixel. We also find certain cases where the alignment of subframes has been
revised for the L3B data, but not for the L1B scenes which still exhibit severe striping (Supplementary Figure S1).

## 3.2 Past approaches to improve scene-to-scene co-registration

Several studies have worked on finding ways to improve the relative geolocation accuracy of PlanetScope scenes. Kääb et al.
(2017) have improved scene-to-scene alignment of PS2 L1B data over stable ground through the fitting of first and second-



order polynomials to remove global shift and rotation (first order) and margin effects appearing in the uncorrected L1B data due to superposition of lens and image distortions (second order). Kääb et al. (2019) used a linear (first-order) polynomial co-registration for PS2 L3B data based on tie points extracted along the shoreline of the studied river. Feng et al. (2019) have also put forward polynomial curve fitting to correct long-wavelength ramp errors in PlanetScope L3B data. Dille et al. (2021), who have targeted a slow-moving landslide, have corrected global shifts by registering orthorectified L3B PlanetScope scenes to a Pléiades orthomosaic using the AROSICS framework (Scheffler et al., 2017). A recent study by Aati et al. (2022a) proposed an unsupervised learning technique to separate true displacement signals from artifacts and noise based on maximum spatiotemporal coherence from a stack of correlations derived from orthorectified L3B mosaic images, applicable to all PlanetScope generations (Aati et al., 2022a). Additionally, Aati et al. (2022a) suggested refining the RFM of L1B data based on pointing error minimization. The relative pointing error is a metric that was first introduced by de Franchis et al. (2014) for the improvement of stereo-reconstruction capabilities of satellite stereo imagery. Given two corresponding points in two images, the pointing error is defined as the distance between a point and the related epipolar curve (or line for frame images) and can be modeled as a simple translation or affine transformation (de Franchis et al., 2014; Aati et al., 2022a). The RFM of a scene can then be refined based on the approximated pointing error. Aati et al. (2022a) used the refined RFMs to construct DEMs out of PS2 scenes and assess elevation changes. But they also showed that PS2 images rectified with refined RPC models exhibited displacements following a zero-centered Gaussian distribution with ∼1 pixel standard deviation when correlated. PS2.SD and PSB.SD scenes were considered unsuitable for stereo-reconstruction due to their strong striping related to the misalignment of sub-frames and other artifacts (Aati et al., 2022a).

In our study, we aim to contribute to existing approaches to enhance the relative geolocation accuracy of PlanetScope scenes, particularly of the modern PSB.SD instruments, which are currently the only sensors providing data. We particularly focus on mitigating the orthorectification error introduced by outdated elevation data. Orthorectification errors result in local misregistration of pixels so they cannot be removed through global shifts or polynomial fitting. Due to its systematic nature, it is also difficult to separate error from true displacement when stacking many disparity grids. To address this issue, we propose a mitigation approach that involves selecting optimal correlation pairs when working with orthorectified L3B data. Additionally, we generate Digital Elevation Models (DEMs) from L1B data that more accurately represent the terrain at the time of image acquisition, enabling improved orthorectification. Through the implementation of these techniques at both processing levels, we are able to achieve sub-pixel accuracy which strongly enhances the quality of offset-tracking results across our test sites.

# 4 Data and Methods

## 4.1 Data selection and preprocessing

In this study, we worked with both orthorectified L3B scenes and raw L1B data. For both study regions, we selected cloud-free images only. We also required scenes to have a minimum overlap of 99% with the area of interest (AOI) and pass all GCP checks imposed by Planet, so that they are geometrically aligned. In the case of L3B data, we took advantage of the option to download pre-clipped scenes that match the AOI. For L1B data, where the clipping option is not available, we downloaded the



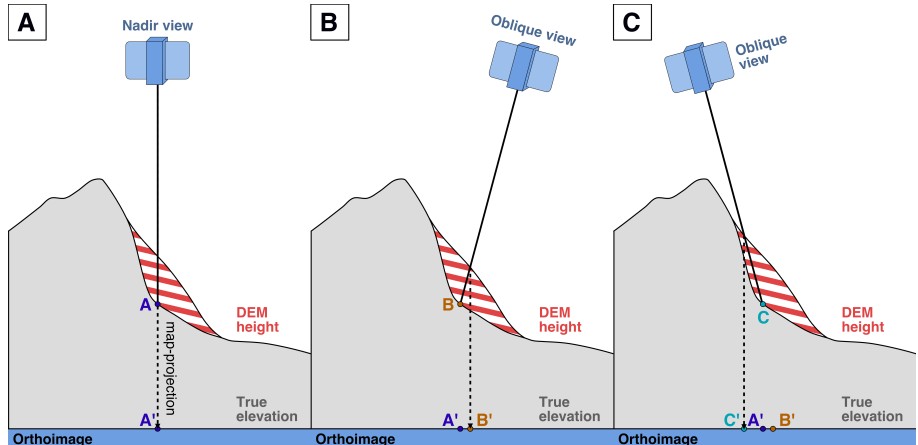

**Figure 4.** Conceptional sketch of the orthorectification error resulting from outdated DEM heights for nadir (A) and oblique views (B, C). When the DEM used to correct for geometric distortions no longer reflects the true topography at the time of image acquisition, e.g. due to landsliding, a common point seen in images A-C may be projected to wrong ground locations. Oblique-view scenes are more susceptible to this effect due to increased geometric distortions that the orthorectification aims to compensate for. Also, the misprojection scales with the elevation difference between present topography and DEM height. The misprojection of a common point leads to an apparent offset signal that interferes with the true displacement related to landslide motion.

full scenes and locally cropped the raw image. Clipping extents were determined by converting the longitude and latitude of the AOI's upper left corner to pixel positions via the RPCs and specifying a clip size in pixels.

The PlanetScope constellation acquires multi-band data, but only a single band is needed for image correlation. Rather than creating a pseudo-panchromatic image from the RGB channels, we chose to work with a single band only due to inter-band
misalignment between spectral bands for PS2.SD and PSB.SD instruments, especially in steep, rugged terrain or across moving objects (Aati et al., 2022a; Planet, 2022a). The green band (band 2) was selected, as it is least affected by atmospheric effects or vegetation. It should be noted that for 8-band PSB.SD data, band 2 corresponds to the blue part of the electromagnetic spectrum, but Planet also offers the option to download a 4-band PSB.SD scenes that correspond to RGB + NIR. The green band was isolated and only the resulting single-band scenes were considered in the subsequent analysis. Detailed information
and code for all data selection and preprocessing steps can be found on GitHub (https://github.com/UP-RS-ESP/PlanetScope_landslide_tracking).

## 4.2   Forming optimal correlation pairs using PlanetScope L3B data

Most users of PlanetScope data will rely on L3B data, where geometric, radiometric, and atmospheric corrections have already
been applied. Additionally, L3B scenes are available in clipped format through the Planet Explorer or Planet Download API, enabling users to maximize the number of scenes that can be downloaded within their monthly quota, particularly for smaller





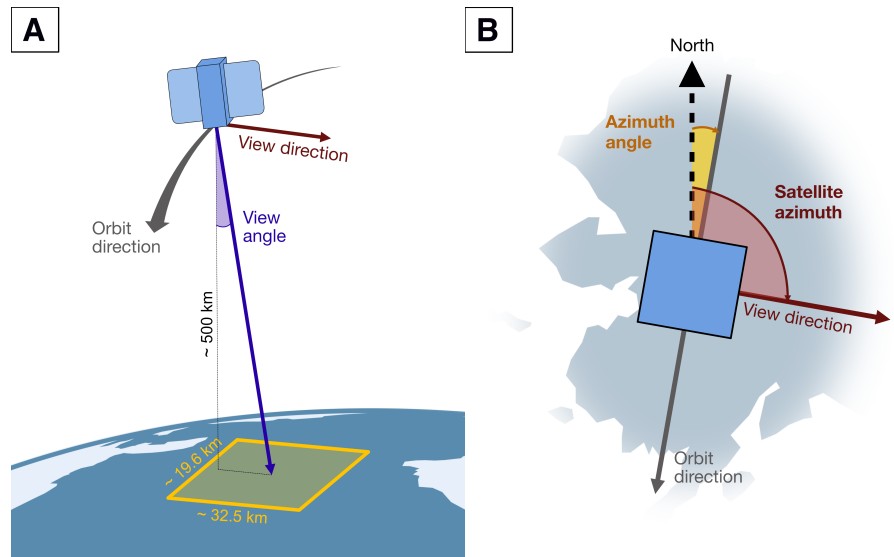

**Figure 5.** Sketch of PlanetScope satellite geometry from side (A) and top-down view (B). For reducing erroneous displacement signals related to orthorectification errors, it is essential to only correlate L3B data that was acquired from similar perspectives. This includes the view angle, which is the satellite's off-nadir viewing angle across track (Planet, 2022b). In the JSON metadata, the view angle is always provided in positive numbers, typically ranging between 0 and 5. The off-nadir angle needs to be considered in combination with the satellite azimuth, which describes the angle between the satellite's view direction and true north (B) and ranges between 0 and 360°. The satellite azimuth thus determines if the satellite is left- or right-looking. Scenes acquired from an opposite view direction at high view angles are strongest affected by orthorectification errors. The satellite azimuth should not be confused with the azimuth angle that is given in the XML metadata of a scene. The azimuth angle describes the angle between true north and the scan line direction.

study areas. Considering these conveniences, we wanted to investigate if and under which conditions L3B is still usable for reliable image offset tracking across deforming terrain, even when there is no control on the DEM used in the orthorectification process. PlanetScope imagery is typically orthorectified using the SRTM DEM (Planet, 2022b), which may be outdated by over two decades depending on the acquisition date of the analyzed imagery. Given the dynamic nature of landslides, it can be expected that terrain changes have occurred since the acquisition of the SRTM DEM unless the landslide motion is very small or purely translational. In the case of the Siguas and Del Medio landslides, a comparison between the SRTM DEM from 2000 (now called NASADEM after reprocessing) (NASA JPL, 2021) and the Copernicus DEM from 2012-2015 (European Space Agency, 2021) reveals elevation changes of more than 75 meters in source and deposition area (Figure 2). These changes introduce lateral distortions that manifest as apparent displacements in the derived disparity maps. These displacements often exceed the ∼ 2-pixel relative geolocation accuracy between two PlanetScope scenes. The reason for this phenomenon is the discrepancy between the terrain seen at the time of acquisition of the satellite imagery and the DEM used for generating the orthoimage. During orthorectification, geometric image distortions in the acquired scene are corrected and the acquired PlanetScope scene is converted from a satellite-perspective view to an orthoimage by projecting the image pixels on pre-existing





terrain. If the elevation model does not reflect the topography at the time of the acquisition of the satellite imagery, pixels capturing the new topography will be misplaced in the orthophoto. The amount to which this misplacement occurs depends on the elevation mismatch between reference DEM and actual topography, as well as the view angle of the satellite at which the imagery was taken – scenes acquired from a nadir view have much lower perspective distortions and will consequently be much less affected than scenes from an oblique view.

The pixel misplacement in a single scene alone, however, does not yet produce an artificial offset signal in the derived disparity maps. The problem arises if a common point of interest is observed from two different perspectives and corrected using an outdated reference surface. In this scenario, the common point will no longer be located at the same elevation on the reference DEM. Consequently, during orthorectification, the point will be mapped to two different positions, introducing an offset signal where it should not exist (Figure 4). This false offset signal will in the following be referred to as orthorectification error.

PlanetScope L3B scenes taken from different angles and corrected with a reference DEM that no longer reflects the current topography will be strongly affected by orthorectification errors that introduce significant bias to any attempt of quantifying landslide movement through optical image correlation.

Orthorectification introduces a nonlinear transformation, making it difficult to model DEM errors after orthorectification. Consequently, correcting erroneous orthoprojections among the L3B scenes becomes infeasible. However, in cases where scenes

are acquired from a similar perspective, the lateral misplacement of pixels is expected to be consistent. When correlating these scenes, the pixels may be wrongly positioned, but in a similar place. By limiting the correlation analysis to scenes captured from a similar viewpoint at similar angles, the influence of DEM errors on the true displacement signal can be minimized. For finding images with a common perspective we considered the acquisition parameters *view_angle* and *satellite_azimuth* provided in the JSON metadata of PlanetScope scenes. The advantage of the JSON metadata is that it can be quickly accessed and

browsed via the Planet API. The XML metadata offers more detailed measurements and additional parameters such as the incidence angle but is an asset that needs to be activated and individually downloaded. The view angle describes the spacecraft's across-track off-nadir viewing angle in degrees (Planet, 2022b). It should be noted, that at the time of writing (July 2023), the view angle in the JSON metadata is always provided in positive numbers, typically ranging between 0 and 5. It needs to be jointly considered with the satellite azimuth to determine the direction in which a satellite is looking. The satellite azimuth as

specified in the PlanetScope JSON metadata describes the spacecraft's off-track pointing direction and is given as an angle to true north (Planet, 2022b). It should not be confused with the azimuth angle (*azimuthAngle*) provided in the PlanetScope XML metadata, which refers to the clockwise angle between the spacecraft's scan line direction direction and true north (Planet, 2022b) (Figure 5 B) and is around 7 or 11 degrees for the downloaded scenes at the Siguas site. The view direction given by the satellite azimuth angle is typically near perpendicular to the azimuth angle (approx. 80 - 90 degrees), but can also be close

to 0 or 360 degrees for scenes with a very low view angle. The satellite azimuth is an essential attribute to consider, as the convergence angle of scenes with opposite satellite azimuths can be up to $\sim 10°$ if view angles are high. These scenes are expected to be particularly vulnerable to the impact of orthorectification errors.

To test this hypothesis, we correlated 17 PlanetScope L3B scenes with variable views and satellite azimuth angles covering the Siguas landslide acquired during a $\sim$ 1.5 months period between July 7, 2022, and August 23, 2022, which resulted in 136 dis-





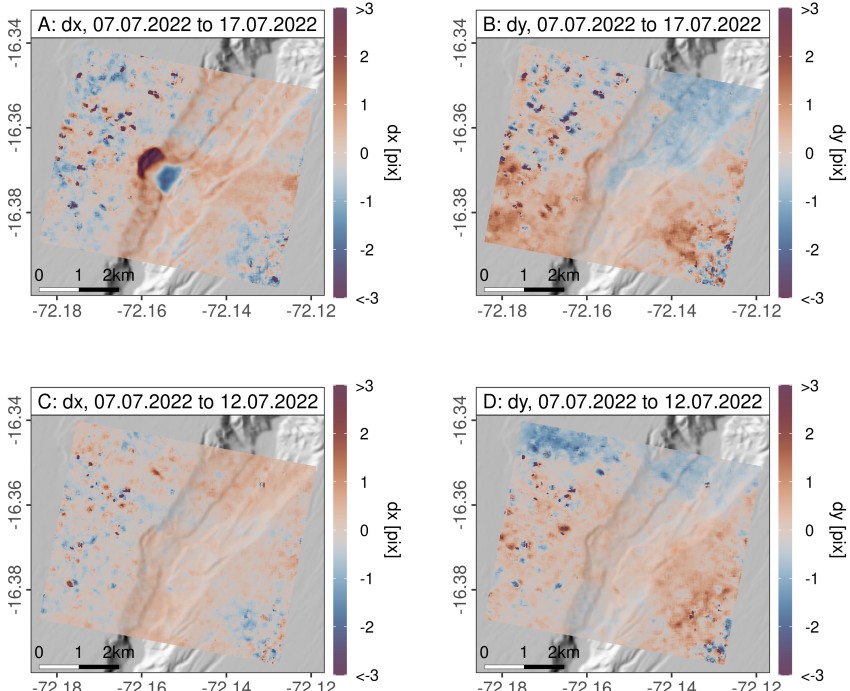

**Figure 6.** Disparity maps generated from two PlanetScope image pairs across the Siguas landslide. Figures A and B show the displacement in EW direction (dx) and NS direction (dy) for PlanetScope scenes acquired between July 7 and 17, 2022 from opposite viewing angles. It is unlikely that the landslide has accumulated displacement above the detection limit throughout this 10-day period, but there is a strong offset signal, particularly in the EW component. This signal correlates with the elevation difference between the topography at the time of image acquisition and the DEM used during the orthorectification process (see Figure 2). The apparent displacement signal disappears when the scene from July 7 is paired with a scene from July 12, 2022 (C and D). In this case, the scenes were acquired with a common satellite azimuth (0.3 $^\circ$ angle difference) and similar view angle (4.1$^\circ$ and 5$^\circ$).

parity maps. A list of all scenes can be found in the Supplementary Material, Table S1. Given that the target is a slow-moving landslide with a velocity of about $\sim$ 10 m/yr, it is unlikely that within that time frame, displacements above the detection limit will have occurred and the landscape can be assumed to be stable. Any offset signal seen in the disparity maps will thus be related to orthorectification errors, other factors limiting the relative geolocation accuracy of PlanetScope scenes as discussed in 3.1, or erroneous matches. As shown in Figure 6 A and B, a prominent displacement signal is observed across the landslide for

an image pair from opposite view angles, particularly in the EW component. However, for other correlation pairs, the landslide appears to be stable over the covered time span (Figure 6 C and D). These scenes are typically characterized by low view angle differences and common satellite azimuths. The dependence of the orthorectification error on the difference between view and satellite azimuth angles is confirmed by all 136 correlation pairs (L3B) formed for the Siguas landslide (Figure 7. Here, we





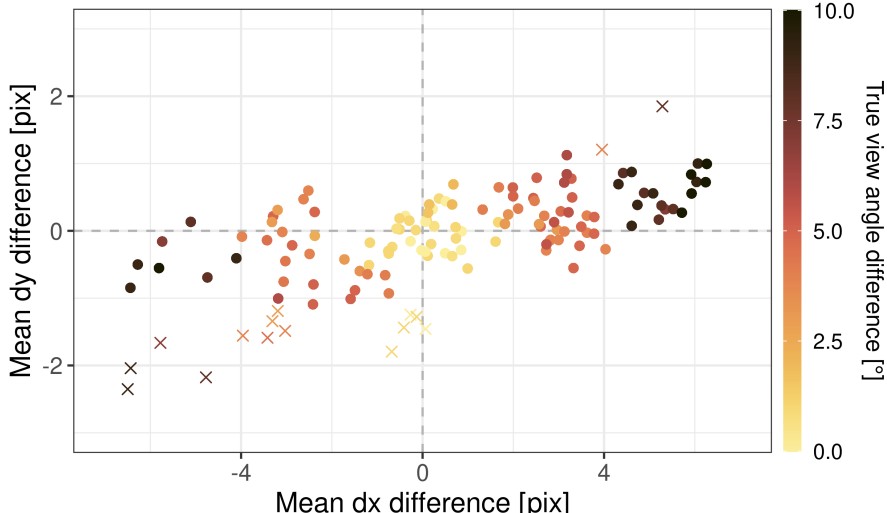

**Figure 7.** Difference of mean displacement in EW (dx) and NS (dy) directions between source and deposition area of the Siguas landslide as outlined in Figure 2 A, derived from 136 correlation pairs of L3B scenes between July 7 and August 23, 2022. The scatter plot shows the degree to which the disparity map is affected by diverging projections related to DEM errors. We generally observe a linear relationship between the strength of the DEM error signal in dx and dy, with dx showing the strongest difference. There are a few data points that deviate from this relationship and exhibit a stronger DEM error signal in the NS component (plotted as crosses instead of points). We found that all of these correlation pairs are related to a single acquisition from August 12, 2022. We assume that this increased signal in dy is related to a slightly different satellite pitch angle, which has not been recorded in the metadata. Colors represent the true view angle difference between two scenes, which considers whether a satellite is left- or right-looking (positive = east, negative = west). We find that orthorectification errors are lowest for scenes taken at common view angles from the same direction. To minimize the bias from orthorectification errors resulting from outdated DEM heights over unstable terrain, only scene pairs with minimal differences in true view angles should be considered for correlation.

extracted the mean displacement within the source and deposition area of the landslide (outlines are shown in Figure 2) and

calculated their difference. If a correlation pair is affected by orthorectification errors, the difference is high, as positive and negative elevation changes will laterally offset the projected pixels in opposite directions. We find that when view angles are considered together with the satellite azimuth to attribute positive and negative signs depending on the cross-track pointing direction (positive = east, negative = west), it becomes clear that the true view angle difference is a robust predictor of how strong a correlation pair will be biased by orthorectification errors (Figure 7).

Based on the true view angle difference, we form groups of PlanetScope L3B scenes acquired between 2020 and 2023 to trace displacement across the Siguas and Del Medio landslides (three groups each). Scenes within a group have a true view angle difference of 0.6° or less. Within these groups, misplacement related to orthorectification error is expected to be consistent and remaining displacement signals can be attributed to landslide motion unless the relative geolocation accuracy is lacking due to other error sources. Correlation pairs are formed among all scenes within a group that have a temporal baseline of six or more





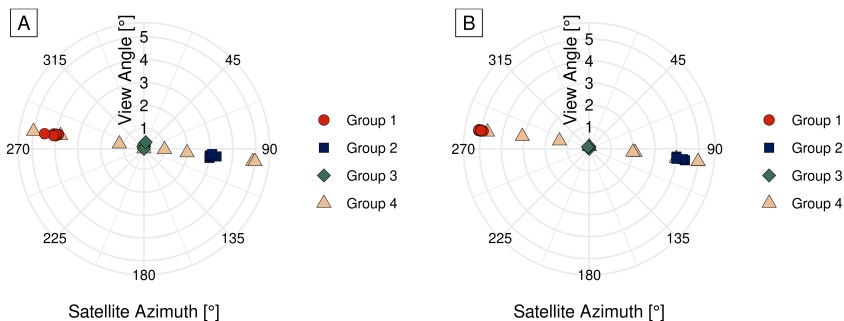

**Figure 8.** Satellite azimuth and view angle of the PlanetScope scenes that were grouped based on their true view angle differences for the Siguas (A) and Del Medio landslides (B). Additionally, we used a group of randomly selected scenes (Group 4) to showcase correlation results as they would be obtained if acquisition parameters are not considered. IDs and acquisition parameters of all scenes are listed in the Supplementary Material, Tables S2-S3.

months (180 days). We impose this limit, to eliminate image pairs where the estimated landslide displacement at the Siguas and Del Medio sites will fall below the detection limit. For comparison, we also select a set of random scenes for a fourth group to compare the derived disparities to a naive selection of PlanetScope scenes with varying view and satellite azimuth angles (Figure 8).

## 4.3 Manual orthorectification of PlanetScope L1B data

When working with the raw L1B data, the user gains control over the choice of reference DEM used for orthorectification. Selecting a reference DEM that closely matches the acquisition time can minimize the impact of DEM errors for scenes acquired from opposite view angles, as the magnitude of the error is influenced by the difference in elevation. In our analysis of the Siguas and Del Medio landslides, we observed a significant reduction in the apparent offset signal in the displacement maps when the raw scenes were projected onto the Copernicus DEM instead of the NASADEM (Figure 9 A-B, D-E). The
orthorectification process was carried out using the *mapproject* tool from Ames Stereo Pipeline (Beyer et al., 2018) with the raw image data, RPCs, and a reference DEM as inputs. Through the use of the Copernicus DEM, the orthorectification error signal was reduced but not eliminated, as the Copernicus DEM is also outdated with respect to the topography seen in the PlanetScope scenes. With no newer high-quality publicly available global elevation data set at hand (Purinton and Bookhagen, 2021), we explored the possibility of generating DEMs from PlanetScope data itself, to more accurately model geometric
distortions in the unrectified scenes. Several authors have shown the potential of exploiting the small B:H ratios of PlanetScope acquisitions with opposite view angles for 3D surface reconstruction (Ghuffar, 2018; Aati and Avouac, 2020; d'Angelo and Reinartz, 2021; Aati et al., 2022a; Huang et al., 2022). However, these studies mostly focused on imagery acquired by the PS2 instruments that are no longer operational, as these scenes (except for the NIR band) represent a single frame in contrast to PSB.SD bands which are composed of individual sub-frames (see Section 3). PSB.SD data were considered to be unsuitable





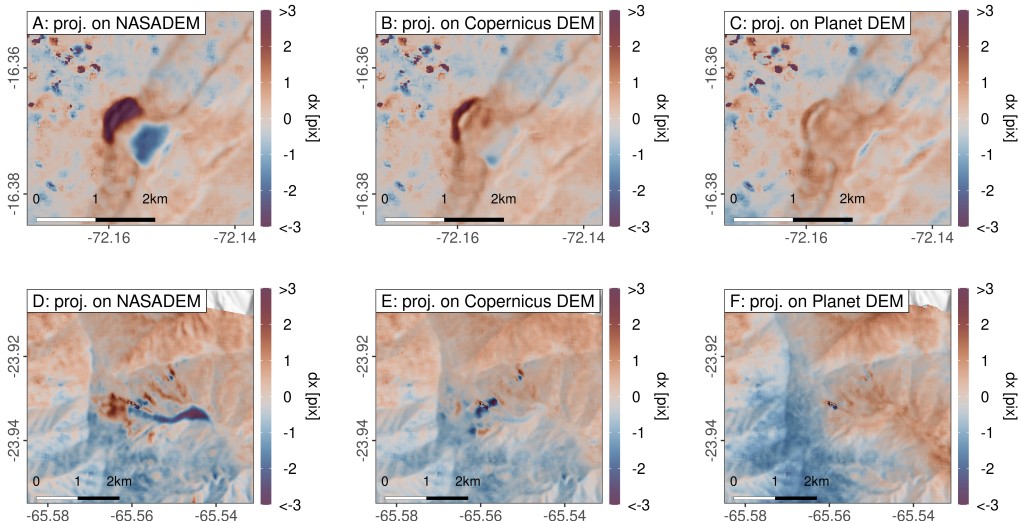

**Figure 9.** Displacement in EW direction estimated across the Siguas (A-C) and Del Medio landslides (D-F) with PlanetScope L1B scenes orthorectified using the NASADEM (A, D), the Copernicus DEM (B, E), and a DEM generated from PlanetScope L1B scenes (C, F). Disparity maps were generated from scene pairs with a minimal temporal baseline: 10 days (07.07.2022 to 17.07.2022) for A-C and 16 days (08.09.2022 to 24.09.2022) for D-F, so the surface can be assumed to be stable. Outdated DEM heights in the reference DEMs produce lateral offset signals in the disparity maps. Projecting PlanetScope scenes onto a newer reference surface (i.e. Copernicus DEM, B and D) reduces but not fully eliminates the apparent displacement related to DEM error. A reference DEM acquired at a similar time as the PlanetScope scenes would be needed in order to fully resolve the DEM-related effects in the derived disparity maps. We suggest that such a surface can be generated from the PlanetScope data itself. Displacements in NS direction (dy) can be found in the Supplementary Material, Figure S2.

for multi-view stereo reconstruction due to their inherent geometric errors which can severely bias the derived surface model (Aati et al., 2022a). However, with the improved sub-frame alignment used for newer PSB.SD scenes and a substantial parallax visible in disparity maps derived L1B data acquired from oblique (5°) view angles and opposite satellite viewing directions, we suggest that it is possible to derive low-resolution stereo DEMs that are sufficiently accurate for orthorectification. We emphasize that our objective is not to generate DEMs that are of comparable quality to other elevation data sets such as the

Copernicus DEM or NASADEM, but allow us to produce a smooth, low-resolution (30 m) representation of the terrain at the time of image acquisition.

### 4.3.1   DEM generation

For DEM generation, we used the tools provided by the Ames Stereo Pipeline (Beyer et al., 2018). For both the Siguas and Del Medio sites, we chose scenes with maximum view angles (5 °), opposite satellite azimuths, and a minimal temporal baseline

as stereo pairs: 02.07.2022 and 06.07.2022 (Siguas) and 07.09.2022 and 12.09.2022 (Del Medio). The full DEM generation workflow comprised the following steps:





1. Images were clipped to a common area using the provided RPCs. Because matching of image pixels along the image margins tends to be less reliable, these are often eroded when the final elevation model is generated. We extracted a region larger than the initial area of interest to ensure full DEM coverage.

2. The re-projection error resulting from the camera position and orientation errors was minimized through bundle adjustment.

3. Through an initial execution of the stereo correlation tool using the unprojected images, we obtained a first-order reference surface. The resulting point cloud image was gridded to a 90 m resolution.

4. Images were map-projected onto the previously generated 90-m DEM. Transferring images to object space increases the chance of finding reliable matches in the subsequent correlation step as images appear more similar and search distances are reduced.

5. The final DEM was obtained through stereo correlation of the map-projected images. For triangulation, pixel correspondences are back-projected into image space. The obtained point cloud was gridded to 30 m spatial resolution

For full details on the processing steps, we refer the reader to the corresponding code on our accompanying GitHub site (https://github.com/UP-RS-ESP/PlanetScope_landslide_tracking) and the Ames Stereo Pipeline Documentation (https://stereopipeline.readthedocs.io/en/latest/index.html) for a detailed description of the individual tools.

Stereo processing parameters were largely left at default settings. For sub-pixel refinement, we used sub-pixel mode 2 (affine adaptive window with Bayes Expectation Maximum weighting) as this results in best quality matches at the expense of longer runtime (Beyer et al., 2018). Correlation kernels were set to 35 x 35 pixels for the Del Medio landslide, 65 x 65 pixels for the Siguas landslide, and + 10 pixels for the sub-pixel kernel. Larger correlation kernels will increase the chance for a pixel to be successfully matched, yet sharp ridge and valley features may be smoothed over (Purinton et al., 2023). At 30 m resolution, the loss of sharpness should be mostly negligible. Our primary concern was to obtain a continuous and artifact-free surface that can be used for improved orthorectification of the L1B data. This task was particularly challenging for the Siguas landslide which is surrounded by flat, low-texture areas that often relate to lower DEM quality (Purinton et al., 2023). To compensate for the low image texture, we utilized a significantly larger correlation kernel size compared to the Del Medio landslide, where the presence of steep and variable terrain provided sufficient image texture for reliable matching even with smaller correlation kernels. Hillshades of the resulting DEMs are displayed in Figure 10 A and D.

### 4.3.2 DEM alignment

The DEMs generated from PlanetScope data had to be aligned to a reference surface because we observed significant lateral shifts and vertical offset with respect to the true geolocation. We initially aligned DEMs for both study sites to the NASADEM (NASA JPL, 2021) using *demcoreg* (Shean et al., 2016) according to the algorithm described by Nuth and Kääb (2011). The alignment worked well for the steep mountainous terrain surrounding the Del Medio landslide but proved to be much more difficult for the low-relief Siguas site. Here, neither the method proposed by Nuth and Kääb (2011), nor other alignment





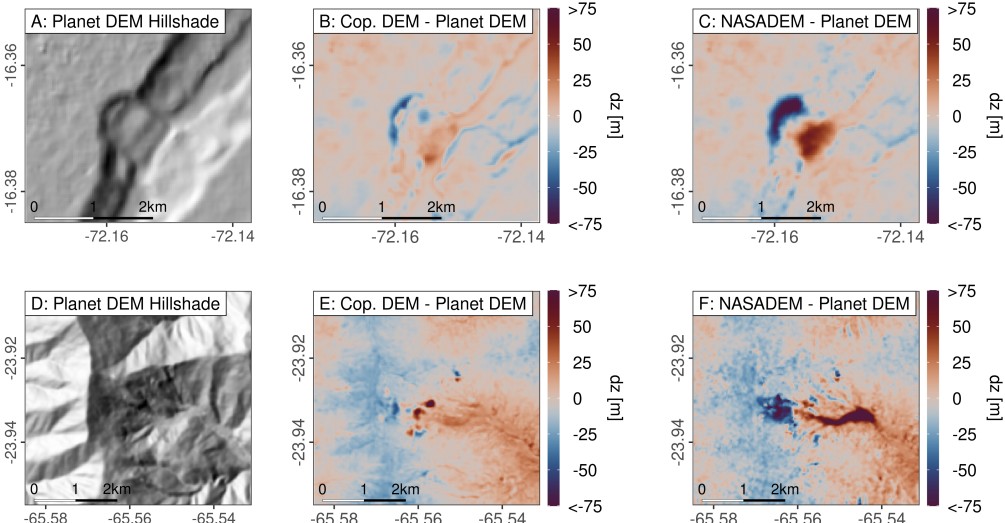

**Figure 10.** A: Hillshade of the DEM generated from two PlanetScope acquisitions (02.07.2022 and 06.07.2022) over the Siguas landslide. The DEM was placed at a position that minimized the disparity between pixel correspondences between two PlanetScope scenes when transferred into object space using the given DEM. Figures B and C show elevation differences between the PlanetScope DEM to the Copernicus DEM and NASADEM which largely reflect the elevation changes that occurred due to landsliding. D: Hillshade of the DEM generated from two PlanetScope acquisitions 07.09.2022 and 12.09.2022 over the Del Medio landslide. Figures E and F show elevation differences between the PlanetScope DEM and the Copernicus DEM and NASADEM.

methods supported by *demcoreg* (sum of absolute differences, normalized cross-correlation), were able to fully eliminate the
shifts between the Planet and NASADEM. Because of these shortcomings, we developed an alternative alignment method that relies on the disparity estimated from a stable (short temporal baseline) PlanetScope image pair acquired from different perspectives (view angles and satellite azimuth): When two unprojected L1B scenes are correlated in image space and the obtained pixel correspondences are transferred into object space using the given RPCs and a reference DEM, the displacement between the projected pixels should ideally be zero. If, however, the reference DEM is shifted, the pixel correspondences
will indicate an offset related to wrong DEM heights. Based on this assumption, we iteratively shift the DEM in both X and Y directions, project pixel correspondences using the new DEM position, and find the location that minimizes the sum of displacement between matching points. The remaining elevation differences between the Copernicus DEM and NASADEM (Figure 10 B-C, E-F) reflect terrain changes due to landsliding and correspond well to the lateral displacement seen in Figure 9. We observe a slight relation between DEM difference and topography, particularly for the Del Medio site. We attribute this
to the low B:H ratio of PlanetScope acquisitions which enables only limited depth perception. While this elevation mismatch will introduce orthorectification errors (see Figure 9 E), the resulting offset signal is systematic and can be compensated for as described in Section 4.5.1.





The final aligned DEMs were used to orthorectify (map-project) all remaining PlanetScope L1B acquisitions resulting in orthorectified scenes (processing level equal to L3B data) that were then correlated among each other to track displacement
across the Siguas and Del Medio landslides.

## 4.4 Image correlation

Image correlation was performed using Ames Stereo Pipeline, but there are other tools for image-offset tracking available, e.g. COSI-Corr (Leprince et al., 2007), geoCosiCorr3D (Aati et al., 2022b), autoRIFT (Gardner et al., 2018; Lei et al., 2021), or MPIC-OPT (Provost et al., 2022). Finding correspondences between two images is one of the key steps in stereo reconstruction.
Since the release of version 3.1.0 (Beyer et al., 2022), Ames Stereo Pipeline has introduced the option to perform stereo correlation in correlator mode, which identifies the disparity between two given scenes without requiring camera information and performing a triangulation of pixel correspondences. We make use of this to compare two PlanetScope scenes and find the disparity between them. Processing parameters were largely left at or close to the default settings. We used Block Matching (BM) as a correlation algorithm with a slightly larger correlation kernel of 35 x 35 pixels and a sub-pixel kernel of 45 x 45
pixels and sub-pixel mode 2 (Bayes EM) for sub-pixel refinement. Block-matching extracts a small window centered around a pixel in the left (reference) image and slides it over the right (secondary) image within a specified search distance. The disparity is determined by minimizing a cost function, which is computed using Normalized Cross-Correlation (Beyer et al., 2018). We experimented with the Ames Stereo Pipeline implementation of the More Global Matching (MGM) algorithm (Facciolo et al., 2015) instead of BM, as it has shown improved matching capabilities, especially in low-texture areas (Purinton et al., 2023),
but we did not recognize significant changes in the derived landslide velocity.

Correlation results comprise a three-band output grid storing shifts in EW direction (positive values indicate movement to the east), displacements in NS direction (positive values indicate movement to the north), and a good pixel map indicating which pixels were successfully matched.

## 4.5 Postprocessing

### 4.5.1 Removal of global shifts, ramp errors, and remaining stereoscopic effects

To correct for global shifts in the displacement maps derived from L3B and orthorectified L1B data, we subtracted the median displacement of both EW and NS components. However, even after this correction, we observed remaining systematic displacement signals at the order of $\pm$ 1-2 pixels in several disparity maps that cannot be compensated for with a simple translation. These signals can be roughly separated into three categories: (1) long-wavelength ramp errors (Feng et al., 2019) with opposite
disparities measures towards the AOIs margins; (2) a topographic signal, particularly in the area of the Del Medio landslide (Figure 11 A,B) which we relate to mismatches in the PlanetScope DEM in case of the orthorectified L1B data, but also RPC inaccuracies; and (3) striping due to misalignment of sub-frames, particularly in the NS component. If needed, these remaining systematic distortions can be compensated for by approximating the estimated disparity with a second-order polynomial fit





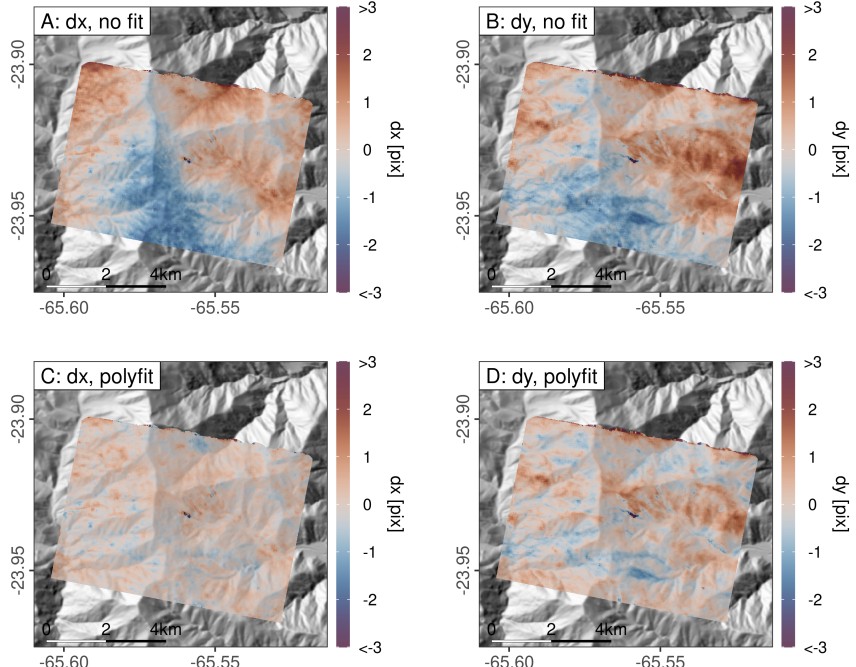

**Figure 11.** Disparity maps estimated across the Del Medio landslide based on an L1B scene pair from 08.09.2022 and 24.09.2022 (same as shown in Figure 9), which was projected onto a DEM generated from PlanetScope data. The uncorrected disparity maps (A,B) still show a subtle topographic signal which corresponds to stereoscopic effects that are not fully removed during orthorectification. While this topographic component may be partially related to the geometry of the DEM generated from PlanetScope data, a similar pattern can also be observed in disparity maps derived from L3B data, pointing to inaccuracies in the RPCs. The remaining parallax signal can be efficiently removed by approximating the derived disparity through a polynomial fit which takes into account local topography. Subtracting this fit results in zero-centered offset maps, which is the expected result for images acquired close in time. An example of the correction of the Siguas site can be found in the Supplementary Material, Figure S3.

including an elevation (topographic) component:

$$dx, dy = aX^2 + bY^2 + cZ^2 + dXY + eXZ + fYZ + gX + hY + iZ + j \tag{1}$$

where dx and dy are the estimated displacements in EW and NS direction, X and Y are the corresponding pixel locations, Z is the elevation and a-j are the coefficients which are estimated using least-square optimization. X, Y, and Z positions were linearly scaled by their maxima between 0 and 1 to ensure common value ranges. The approximated disparity is then subtracted from the original disparity, eliminating topographic signals and ramp effects (Figure 11). To reduce the impact of outliers and noise on the polynomial fit, we excluded displacements below and above the 5th and 95th percentiles disparity. Alternatively, offsets over the landslide could be removed by applying an external landslide mask before the least-squared optimization step. To find



an appropriate fit, it is essential that the displacement maps cover a substantial portion of terrain surrounding the landslide, so that the majority of the scene contains stable ground.

#### 4.5.2 Velocity estimation

Displacement in X and Y directions as obtained from the stereo correlation can be translated to annual velocity by calculating the magnitude of the offset and converting it into meters per year:

$$v = \frac{\sqrt{(dx^2 + dy^2)} * r}{dt} * 365 \tag{2}$$

where v is velocity in m/yr, dx and dy are the estimated disparities in X (EW) and Y (NS) direction in pixel units, r is the raster resolution (3 m), and dt is the temporal baseline between the correlated images in days. To show the average landslide

movement over the entire monitoring period, we stacked all velocity grids within a group and calculated mean and standard deviation for each pixel. For obtaining a velocity timeline, we only considered only pixels inside the area of the landslide which we manually outlined. From all velocity grids, we extracted pixels inside the landslide mask and calculated the mean and standard deviation.

## 5 Results

### 5.1 L3B data

For PlanetScope L3B data, we find that a careful selection of correlation pairs based on common perspectives (similar view and satellite azimuth angles) can substantially reduce erroneous displacement signals related to the use of outdated DEMs during orthorectification. Figure 12 shows the standard deviation of displacement velocity and direction calculated from all L3B data correlation pairs in a group for the Siguas (A-H) and Del Medio (I-P) test sites. We observe that groups 1-3, which

have been selected based on similar view angles, show a very similar displacement signal across the landslide surface. This is indicated by low standard deviations in estimated velocities and directions throughout the correlation pairs. Slightly higher standard deviations at the landslide toe are likely related to transient changes in velocity. In contrast, when PlanetScope scenes are selected randomly (group 4), variations in view angle and satellite azimuth, combined with outdated DEM surfaces used for orthorectification, result in misprojections over the landslide surface. These misprojections introduce biases in velocity

estimates, as artificial offset signals mix with true displacements. When averaged, velocity measurements from group 4 are overall higher and show a high standard deviation in areas where elevation changes have occurred since the acquisition of the SRTM DEM (Figure 10). Orthorectification errors also add a bias to the estimated motion trajectories. While the displacement vector fields derived from correlation pairs in groups 1-3 show a coherent and downhill motion, average results from group 4 suggest a trajectory parallel or even opposite to the steepest descent (Supplementary Figures S4 and S5). We emphasize

that the results presented in these figures represent average offset estimates obtained from 15 to 33 stacked disparity maps per group. Individual correlation results from group 4 exhibit even more unrealistic motion patterns, depending on the severity



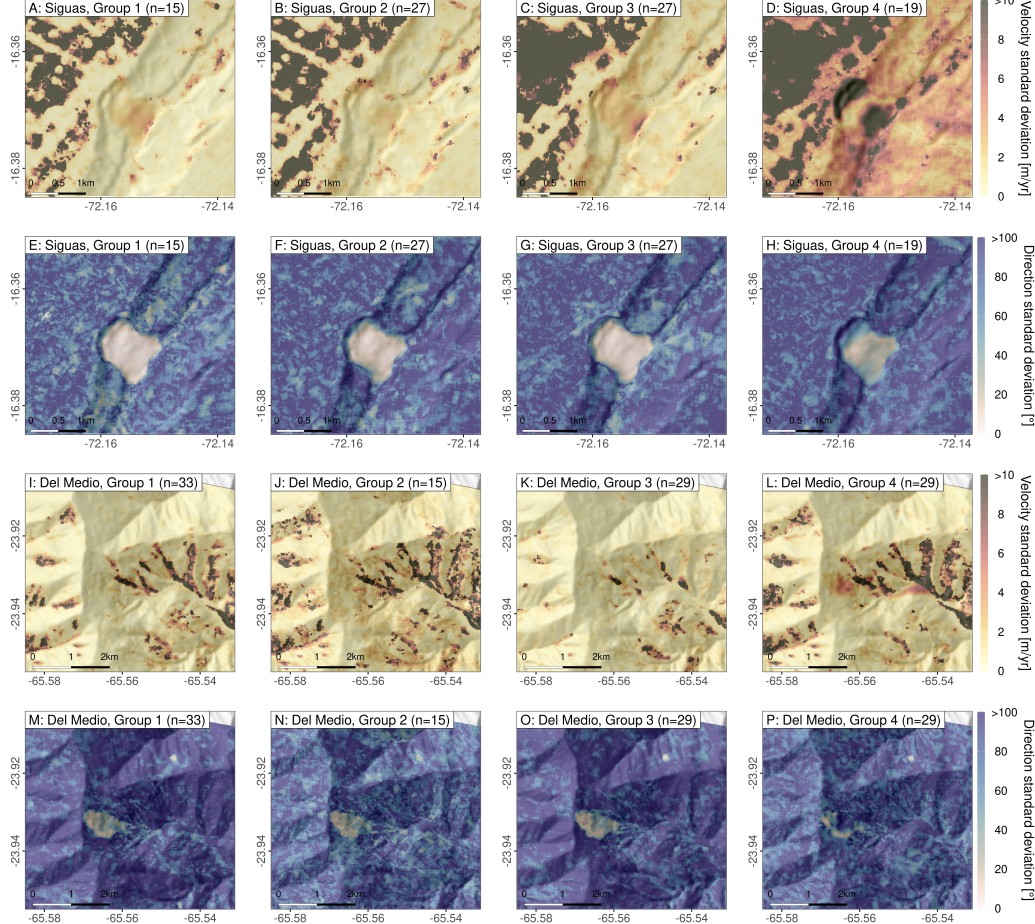

**Figure 12.** Standard deviation of displacement velocity and direction within each L3B data correlation group for the Siguas (A-H) and Del Medio (I-P) landslides. High standard deviations indicate that over a certain area, measurements are highly variable. This is the case for the agricultural areas (Siguas) and shadowed areas in steep terrain (Del Medio) where land cover and variable shadows cause erroneous matches. Displacement directions show high standard deviations throughout stable terrain, where low-magnitude displacement vectors will point in variable directions. Across the landslide surface, however, pixels will move coherently into a common direction resulting in low standard deviations. For both test sites, groups 1-3, which were selected based on common view and satellite azimuth angles, estimates of both the velocity and direction of landslide motion are largely coherent. Slightly higher standard deviations across the landslide surface (A-C) can be explained through transient changes in landslide velocity. For the randomly selected scenes (group 4), velocity and direction estimates are much more variable and show a strong correlation with elevation changes that have occurred since the acquisition of the SRTM DEM (Figure 10). This is a strong indication that disparity maps are biased by orthorectification errors which will mix with the true displacement signal. Correlation results from naively chosen PlanetScope L3B scenes suggest artificially elevated velocities across areas with changing terrain and wrong motion trajectories (see vector plots in the Supplementary Material, Figures S4 and S5).



of orthorectification errors. The high variability among individual disparity maps may lead to erroneous assumptions of a more dynamic landslide movement. Conversely, carefully selected correlation pairs enable a detailed investigation of the true temporal variability, as each correlation pair can be considered individually due to their high quality.

## 5.2   L1B data

Disparity maps derived from PlanetScope scenes captured from different perspectives are particularly vulnerable to errors related to outdated DEM heights. For retrieving unbiased velocity estimates, it is essential to minimize the misprojection error introduced during orthorectification by using a reference surface that reflects the topography at or close to the time of image acquisition to correct for geometric distortions. We find that orthoprojecting L1B data onto a DEM derived from PlanetScope data itself, greatly reduces the artificial displacement signal related to orthorectification errors. The reference DEM, however, needs to be smooth, void-free, and well-aligned with the PlanetScope data. Figure 13 displays the standard deviation of velocity, standard deviation of direction, and average velocity derived from all correlation pairs in groups 4 (variable view and satellite azimuth angles) for the Siguas (A-C) and Del Medio (D-F) landslides when using orthorectified L1B data. Through the use of a reference surface that much more closely resembles the topography at time of acquisition, common points in two images are projected to the same location when orthorectified. Therefore, the orthorectification error signal that appeared in group 4 at the L3B level (Figure 12 D,H,L,P), is drastically reduced. Both estimated velocities and movement directions are much more coherent (low standard deviation) and resemble the measurements retrieved within groups 1-3. Also, the derived displacement fields show a much more realistic, downhill motion compared to the L3B data (Supplementary Figures S4 and S5).

### 5.3   Optimization through a polynomial fit

Figure 14 presents a histogram of EW and NS displacements computed over the entire AOI for the Siguas and Del Medio sites, primarily encompassing stable terrain. We compare the disparity maps obtained from the same image pair using original L3B data, corrected L3B data, and L1B data orthorectified based on a DEM generated from PlanetScope data and further corrected using a polynomial fit. The depicted results are derived from two representative image pairs, one from July 2022 for Siguas and another from September 2022 for Del Medio, but similar outcomes are observed for other correlation pairs.

The use of a polynomial fit to model systematic topographic and ramp errors in the disparity maps significantly lowers the estimated displacement across the majority of the scene (stable terrain). For both test sites, we observe a zero-centered distribution with a spread that is in the sub-pixel range. Uncorrected L3B data available from the Planet Explorer in comparison exhibit higher and more variable disparities within the bounds of the $< 10$ m relative geolocation accuracy. Since we only apply the polynomial fit to the disparity maps and not the PlanetScope imagery itself, we only indirectly improve the co-registration among scenes. In order to directly improve the geolocation accuracy between two PlanetScope scenes, the obtained polynomial fit can, however, be used to remap the secondary image to the reference.

The displacement estimated across the stable surrounding of the landslide is an important indicator of the uncertainty associated with the estimated displacements, as the offset across stable terrain should be zero. Through centering the overall displacement at zero pixels and narrowing the distribution to $\sim 0.5$ pixel interquartile range (IQR), we reduce noise and improve the detection





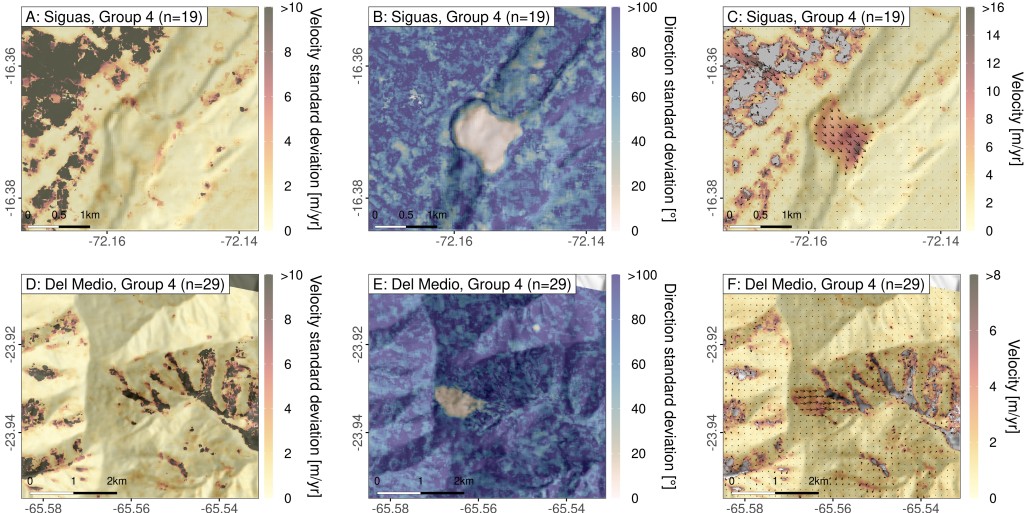

**Figure 13.** Standard deviation of displacement velocity, standard deviation of direction, and mean velocity as vector plots derived from correlation pairs of orthorectified L1B data. We chose the same scenes as in group 4 which comprised randomly selected PlanetScope scenes with variable view and satellite azimuth angles. In contrast to Figure 12, which showed average measurements derived from L3B data, we find that orthorectification errors are minimized when the same scenes are downloaded in L1B format and orthorectified using a smooth, low-resolution reference DEM generated from PlanetScope data itself: velocity and direction estimates are much more consistent among correlation pairs and the vector plot indicates a realistic downhill motion.

of slow motions and measurement accuracy. A correction is therefore essential to make use of the full potential of PlanetScope data for offset tracking. We do not observe significant offset differences for stable terrain between L3B and orthorectified and corrected L1B scenes. The presented approach to mitigate the orthorectification errors is most important for the landslide area, which only constitutes a small portion of the scene. The use of a reference DEM that closely resembles the topography during image acquisition, however, remains essential to improve matching accuracy across unstable terrain when images are acquired
from different angles and directions (Supplementary Figures S6 and S7).

## 5.4 Landslide dynamics

The disparity maps derived from PlanetScope imagery within each correlation group provide valuable insights into the dynamics of the Siguas and Del Medio landslides between 2020 and 2023. Figure 15 illustrates averaged velocities within the landslide boundaries for each correlation pair. For the Siguas landslide, we separated measurements close to the landslide head
scarp from those at the landslide toe, as we found that the velocities vary spatially. The timeline indicates that the Siguas site in 2020 exhibited average velocities exceeding 10 m/yr at the head of the landslide and 15 m/yr at the landslide toe. Velocities gradually decreased to approximately 5-7 m/yr by 2023. These observations align with the self-entrainment process at the Siguas landslide described by Lacroix et al. (2019). The authors suggested that the landslide dynamics are controlled by



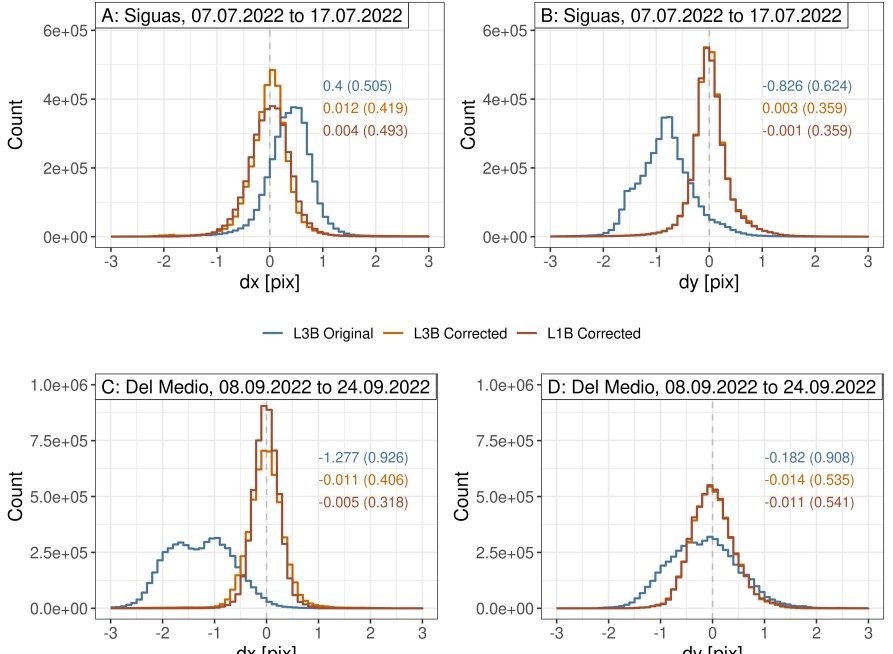

**Figure 14.** Histograms of displacement values in the EW (dx) and NS (dy) directions across the entire study area at the Siguas (A, B) and Del Medio (C, D) sites (refer to Figure 1 for geographic extent) from two representative image pairs (07.07.2022 to 17.07.2022 (Siguas) and 08.09.2022 to 24.09.2022 (Del Medio)). The comparison includes disparity values obtained from Level 3B scenes as downloaded from the Planet Explorer (blue), after applying a polynomial fit to correct for margin and stereoscopic effects (orange), and displacement derived from orthorectified L1B scenes using a DEM generated from PlanetScope data, with remaining distortions corrected using a polynomial fit (red). Colored labels correspond to median and IQR of distribution. Through the correction steps proposed in this study, the co-registration accuracy between two PlanetScope scenes can be lowered to the sub-pixel range over stable terrain, which improves the differentiation of slow landslide motions from noise. Through projecting raw L1B data onto a DEM that more closely aligns with the observed topography, orthorectification errors in the area of the landslide can be significantly reduced, which however, do not strongly show in the presented histogram, as the majority of the investigated terrain is stable. Map views of the plotted distribution can be found in the Supplementary Material, Figures S6 and S7.

sediment supply resulting from the retrogressive motion of the head scarp. Following this, we assume that the higher velocities in 2020 are related to a failure at the head scarp that caused an initial acceleration of the landslide body below and which then gradually decayed. In contrast, the Del Medio landslide maintained a relatively constant and spatially coherent motion with average velocities ranging between 2-4 m/yr between 2020 and 2023.



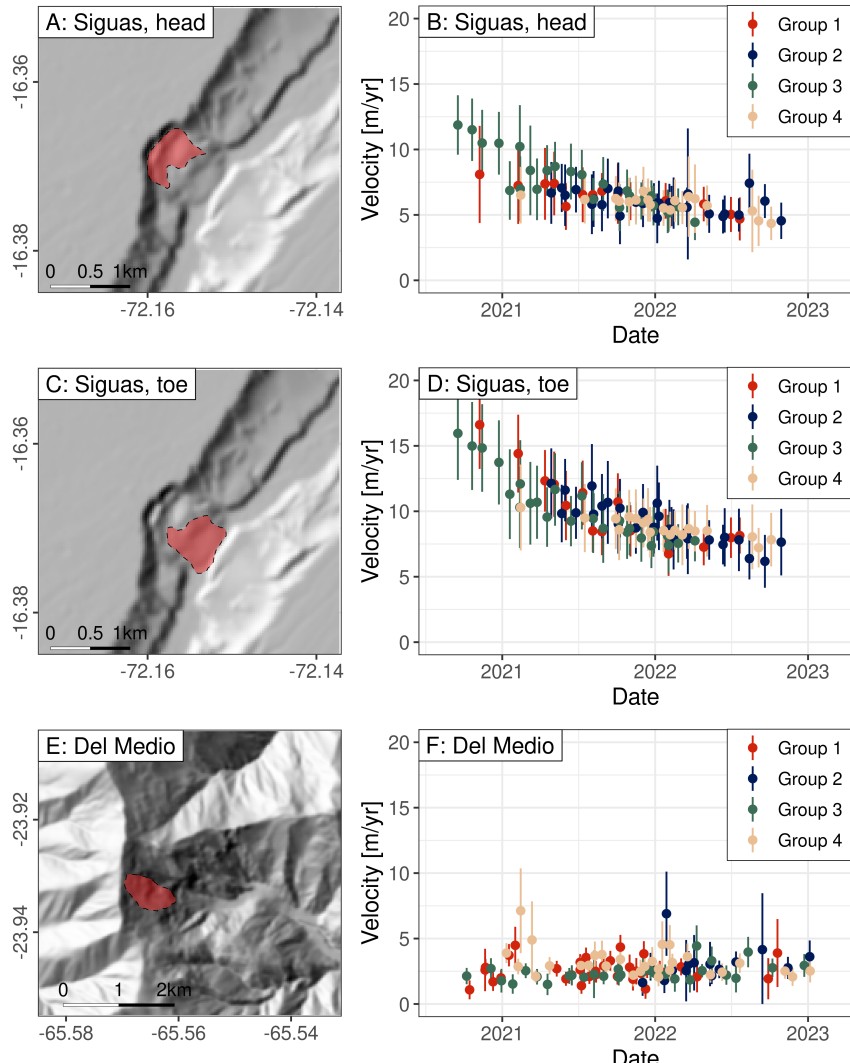

**Figure 15.** Mean velocities and standard deviation of velocities extracted at the Siguas (A-D) and Del Medio (E-F) landslides within the red outlines shown to the right for every correlation pair. Velocity measurements are plotted at the middle of the temporal baseline between the reference and secondary acquisitions. Colors represent correlation groups of L3B data with common view and satellite azimuth angles (groups 1-3). We also included group 4 but used the measurements from the orthorectified L1B data, as L3B data are biased by orthorectification errors and are not useable in this context. For Siguas, we separately considered the landslide head and toe region, as estimated displacements vary spatially. The timeline reveals average velocities of 5-12 m/yr close to the head scarp of the Siguas landslide, while the landslide toe moved considerably faster with 7-16 m/yr between 2020 and 2023. Both regions exhibit declining velocities since 2020. The Del Medio landslide in contrast moves as a coherent block at lower velocities of 2-5 m/yr.





## 6   Discussion

### 6.1   Validity of derived velocity timelines

It is important to note that the derived velocity measurements only provide a generalized estimate. Offset estimates obtained from the cross-correlation of two images captured at different points in time represent an average motion over that time span. The actual displacement may occur more abruptly, potentially in pulses controlled by external forcings such as water infiltration, rainstorm events, and earthquakes (e.g., Keefer, 2002; Hilley et al., 2004; Lacroix et al., 2015; Handwerger et al., 2019, 2022). The 3 m spatial and daily temporal resolution of PlanetScope data enables to better capture the spatial and temporal variability of landslide motion in this context. For a detailed assessment of landslide dynamics at the individual test sites, a denser and longer timeline would be needed which is beyond the scope of this work.

### 6.2   Challenges and limitations

#### 6.2.1   Field of view

For the selection of optimal L3B correlation pairs, we have considered view angles and the satellite azimuth only. For frame images with a larger footprint, an additional factor to take into account is the local incidence angle at the location of the landslide. This angle can vary depending on whether the landslide is situated close to the center or margins of a scene. To investigate the potential impact of the local incidence angle, we compared the orthorectification error signal, as illustrated in Figure 6, with the overlap between the reference and secondary images, but found no observable correlation (Supplementary Figure S8). The overlap serves as a rough indicator of whether the landslide occupies a similar position in both acquisitions. Based on this finding, we argue that, given the flight altitude of approximately 500 km and the relatively small scene size of 32.5 x 19.6 km, the variation in incidence angle within a scene can be safely neglected for the current constellation.

#### 6.2.2   Sub-frame misalignment

While we accounted for DEM errors, global shifts, and margin effects through the polynomial fit, one source of distortion that remains challenging to fully address is the misalignment of sub-frames in PSB.SD scenes. This is particularly problematic when multiple sub-frames are misaligned, causing variable stripes throughout the scene. Single stripe misalignments as shown in Figure 3 C on the other hand can be compensated quite well through the polynomial fit. Although image processing by Planet seems to have made improvements to the pipeline for PSB.SD scene composition in more recent acquisitions, slight striping is still visible in some disparity maps even after correction, particularly in the NS component (Figure 11 D). Without access to the original L1A data, it is difficult to correct for these effects, because the GCPs and homography that went into the alignment process are unknown and impossible to reconstruct. As such, the current option is to acknowledge the inherent limitations and tolerate a marginally lower accuracy in the NS component.




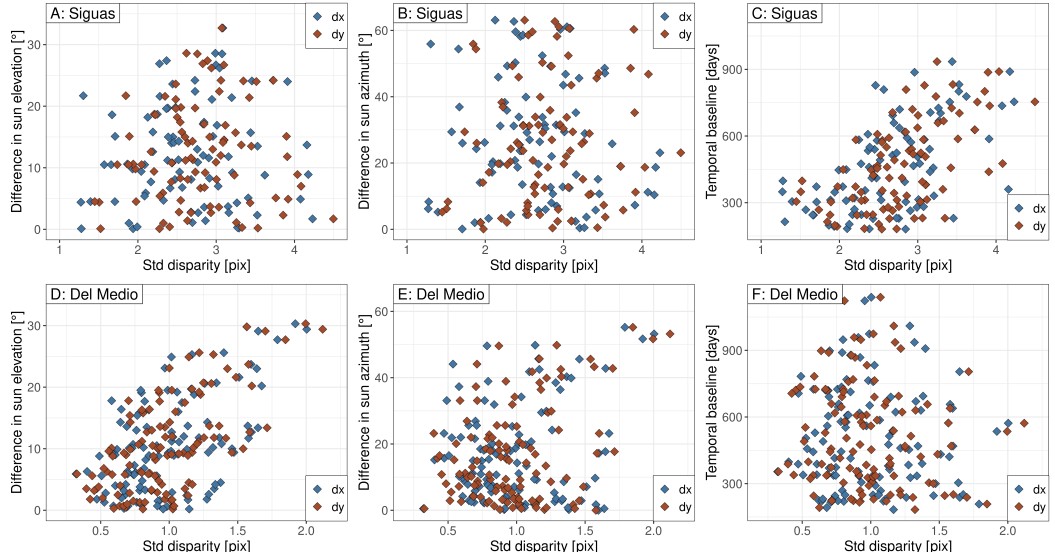

**Figure 16.** Standard deviation of estimated displacement across the entire disparity map for all L3B correlation pairs at the Siguas (A-C, n = 88) and Del Medio (D-F, n = 114) test sites, plotted against differences in sun elevation angles, sun azimuth angles, and temporal baselines between reference and secondary images. We observe higher standard deviations at the Siguas site (note the different x-axes compared to Del Medio) which can be attributed to the agricultural areas towards the northwest of the AOI (see Figure 12). An increasing time difference between two acquisitions leads to these highly dynamic areas becoming more and more dissimilar which in turn results in more mismatches and a higher standard deviation (C). In mountainous terrain disparities exhibit a stronger dependence on sun position, particularly the sun elevation angle (D), due to varying illumination conditions that generate different-looking shadows, often resulting in an apparent elevated displacement signal.

### 6.2.3 Variable shading and changing landcover

Illumination conditions, which can vary significantly between different seasons, pose a well-known challenge for image cross-correlation, especially in rugged mountainous terrain (e.g., Lacroix et al., 2015). Variable shadows across the scene can result in wrong matches that introduce erroneous lateral offsets in the estimated disparity grid. Similarly, rapid land cover changes, e.g., due to farming or seasonal vegetation changes complicate the matching process because the two scenes become increasingly dissimilar. Figure 16 shows the influence of different illumination conditions (difference in sun elevation and sun azimuth an-

gle), as well as the temporal difference between the acquisition of the reference and secondary image on the standard deviation of derived displacement in EW and NS directions. In the case of the Siguas site (Figure 16 A-C), we find overall higher standard deviations which can be attributed to the agricultural areas in the northwest of the study area, which undergo frequent changes in appearance due to crop growth and harvesting. In these areas, the correlation engine often fails to find correspondences between images or produces incorrect matches, leading to false displacement signals. Consequently, we find the strongest re-

lationship between displacement variability and temporal baseline of the acquisition pairs. The more time has passed between




the state that is captured by reference and secondary images, the more changes have occurred and the more difficult it will be to match them. The influence of the changing farmland overshadows any effect of variable illumination conditions which is comparably small because the Siguas site has low relief. In contrast, the mountainous terrain of the Del Medio catchment is characterized by steep slopes and narrow ridges that throw different-looking shadows on the landscape which complicates the matching process. The compromised matching capabilities impact the overall standard deviation estimated across the disparity map, with higher disparities observed when illumination conditions differ. The difference in sun elevation angle hereby seems to have a stronger impact compared to the difference in sun azimuth angles (Figure 16 D-E).

The presence of changing land cover or shaded areas may negatively impact the correction capacities of the polynomial fit if these are not filtered properly, as the fit relies on matching pixels found across the entire study area. We recommend choosing a study area that covers mostly stable terrain with relatively steady land cover. Shadows and changing land cover can further introduce high uncertainties when affecting the landslide surface. In such cases, only scenes with a short temporal baseline and/or from similar seasons should be considered for correlation. In the case of the Siguas and Del Medio test sides, the landslide surface is mostly shade- and vegetation free, and we can assume that the velocity measurements over the landslide body are not compromised by poor matching capabilities.

## 6.3 Transferability to other regions and targets

The proposed approaches are generally transferable to other study sites and moving targets. We provide a list of items to consider:

1. *Vegetated terrain:* matching capabilities are expected to decrease in areas with dense or seasonally variable vegetation cover, as large dissimilarities between reference and secondary scenes will cause the correlation to fail or produce false matches. Here, only scenes with a short temporal baseline may be eligible for matching, which in turn requires the target to have moved more than the detection limit during this time. If this is not the case, cross-correlation based on optical imagery is likely not the method of choice.

2. *Rapid events:* catastrophic landslide failures or co-seismic displacement may cause significant topographic changes that in turn result in orthorectification errors if scenes acquired from different view and satellite azimuth angles are correlated. As this work suggests, this bias can be avoided if correlation pairs are limited to scenes acquired from similar perspectives. If scenes with opposite view angles acquired pre- and post-event need to be compared, the user should generate a pre- and post-event DEM as described in Section 4.3.1 to ensure correct orthoprojection. Similarly, if landslides move at speeds exceeding a few meters per year or are monitored over longer time periods, the cumulative elevation changes may at some point measurably affect disparity maps. In such a case, it may be useful to periodically update the reference DEM and project reference and secondary images onto different, well-aligned DEMs so that the orthorectification error is minimized.

3. *Large study areas:* the extent of a single PlanetScope scene together with the need for sufficient overlap between reference and secondary images restricts the size of targets that can be studied. PSB.SD scenes have a larger footprint (∼





637 km$^2$) than PS2 scenes ($\sim$ 192 km$^2$) Planet (2022b). Regional-scale study areas, however, will require mosaicing

of several subsequent PlanetScope scenes. Planet offers the user to generate custom composites to cover larger areas through the Planet Explorer. Correlation results based on these data, however, reveal artificial displacements at the order of 2 pixels along the margins of the stitched scenes (see Supplementary Figure S9). For tracking motions below this margin-displacement signal, we recommend not to use the currently available composite products and to separately carry out alignment and correction steps to refine the co-registration of subsequent scenes.

## 590   7  Conclusions

PlanetScope data with daily temporal and 3-m spatial resolution hold great potential for capturing and analyzing surface displacements based on optical image correlation. Landslides, however, are particularly susceptible to orthorectification errors, leading to local misregistrations between subsequent scenes which introduces severe bias in disparity estimates. To mitigate orthorectification errors, this study proposes the following approaches:

1. For orthorectified L3B data we suggest that only image pairs acquired from common perspectives should be considered for correlation. It is important to consider both the view angle and the satellite azimuth angle, as provided in the scene's metadata, to determine the satellite's viewing direction. By selecting correlation pairs taken from a common perspective, the misprojections introduced by outdated DEM heights during orthorectification remain consistent and do not manifest as apparent displacement in the disparity maps. This approach limits the number of eligible correlation pairs but yields

reliable and coherent offset estimates.

  2. For unprojected L1B data, the orthorectification error can be reduced even for correlation pairs with opposite view angles by orthorectifying the PlanetScope data based on a DEM that more closely reflects the topography at the time of image acquisition. When no external elevation dataset is available, we suggest exploiting the stereo capabilities of PlanetScope scenes with large convergence angles and short temporal baselines to generate a low-resolution reference DEM.

Additional factors compromising the relative geolocation accuracy, such as the ramp errors and stereoscopic effects that are not fully removed during orthorectification can be efficiently modeled through a polynomial fit that takes into account X and Y position and local elevations. Considering these mitigation strategies, we are able to avoid bias from orthorectification errors and reduce the estimated displacement across stable terrain to zero pixels and $\sim$ 0.5 pixel IQR. Improved co-registration between image pairs yields higher quality disparity maps that allow a more detailed study of the dynamic of landslide processes.

At the Siguas test site, we observe temporally and spatially variable velocities which have declined from 10 - 15 m/yr to 5-7 m/yr between 2020 and 2023. The Del Medio landslide has exhibited a more uniform motion pattern with velocities between 2-5 m/yr between 2020 and 2023.

The observations and processing strategies developed based on these two test sites can be transferred to other regions and dynamic targets, increasing the application potential for studying surface displacement with PlanetScope data to monitor Earth

surface processes.



*Code availability.* The full Python code for processing and correcting PlanetScope data as conducted in this study can be found on GitHub: https://github.com/UP-RS-ESP/PlanetScope_landslide_tracking.

*Author contributions.* Conceptualization, A.M. and B.B.; methodology, A.M. and B.B.; software, A.M.; formal analysis and investigation, A.M.; resources, B.B.; data curation, A.M.; writing - original draft preparation, A.M.; writing - review and editing, A.M and B.B.; visualiza-

tion, A.M and B.B. All authors have read and agreed to the published version of the manuscript.

*Competing interests.* The authors declare that no competing interests are present.

*Acknowledgements.* We acknowledge Planet Labs PBC for granting access to the PlanetScope data through the Research and Education Program (Planet Team, 2022) and thank the Planet Support Team for answering technical questions. We thank the members of the Geological Remote Sensing Group at the University of Potsdam and everyone who contributed to the improvement of this work with their feedback and

suggestions. Scientific color maps used throughout this paper are from Crameri et al. (2020).



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
