# Peer review of "PlanetScope data"

_EGUsphere, 2023_

## Referee Comment (RC2)

**Review of *"Tracking slow-moving landslides with PlanetScope data: new perspectives on the satellite's perspective*" by Mueting and Bookhagen**

Mueting and Bookhagen evaluate the geometric quality of PlanetScope SuperDove data for mapping ground surface displacement, with a particular focus on slow moving landslides. They also propose strategies to select which data to process for best results, and propose a method to rectify image pairs acquired from different orbits to improve ground displacement mapping results. The evaluation is timely, and will be useful for the community as they use PlanetScope imagery in their applications. The manuscript experiments are generally very sound, and I agree with reviewer 1 that the results should definitely be published.

As reviewer 1 rightly pointed out, the manuscript would benefit from a slight restructure. I understand presenting so many sophisticated and novel results as obtained in this study in a succinct way is challenging, so hopefully the below suggestions will be helpful. I enlist below some major and minor suggestions/questions. Please reach out to me if you have any questions/confusions.

**Major points**

- I believe early on in the manuscript, it should be mentioned that the analysis is focussing on PlanetScope SuperDove data. Then the authors can potentially consider reducing the information presented on the earlier generation of PS constellation (current section 3.), and condensing the information on previous efforts which have tried to work with older PS data (Section 3.2). Mentioning some of the previous approaches is valuable, but I do not think a detailed description of those is required. More comments related to this point are also provided in other comments below.
- Consider reducing the dense text on the background of the two landslides. I agree the event description is important, but describing the general characteristics of the events and then pointing the users to published papers for more in-depth details will work better here, as the current paper does not focus on the science of landslides perse.
- Section 3.1 is again pretty dense. One option could be to break into 2 separate sections. The first could be renamed as section 3.1: Expected relative geolocation accuracy, which succinctly describes the geolocation accuracy which is expected in Planet data, quoting numbers from the Planet documentation and previous studies. The second section could describe in more detail the spatial pattern of the typical relative geolocation offset, leading up with the figure 3, describing the four main types of errors. I think there are 4 main issues shown in figure 3 are:

- ○ Error due to **dynamic/outdated topography** in displacement maps obtained from image pairs acquired from different orbits
- ○ Error due to **stereoscopic affects** in the y-direction (again more for pairs collected from different orbits?)
- ○ A **general global shift** which can be corrected by removing the median x and y shifts over static surfaces (this could be present in image pairs collected by both the same and different orbits)
- ○ **Striping effects** in the y-direction due to sub-frame misalignment (will be present in both L1B and L3B data, and potentially for image pairs acquired from both similar and different orbits).

Based on the order you chose to introduce these errors, have small subsections or bullets for these 4 points in the new section. It will then be very clear to the reader that these are the four errors that the authors are going to tackle in the manuscript.

Then, maintain this order when you propose corrections, describe results, and conduct discussions.

There should also be a clear distinction between which correction corrects for what error. So maybe we the corrections are introduced, their section headers could contain information about which of the four errors is being corrected?

Along the same lines, it would be useful to plot the image acquisition geometry skyplot in the third column for each of the two pairs in Figure 3 (as in figure 8), so that it helps us in bringing out the effect of acquisition geometry on some of these errors?

- Current section 4.2 is very long, and the order in which information is presented can be improved. You already talked about how outdated DEM affects displacement mapping in the current section 3.1, ideally the your conceptual figure and text belong there, and not in the methods.
- Similarly, the authors talk about the discussion of using data from a .json file or the scene metadata.xml. All of this is important to consider, but it disrupts the scientific/methodological flow of the paper. Maybe some of this could be transferred from the manuscript to the github repository readme or something?
- DEM generation section have no mention of how accurate the output DEM is? I understand the authors want an update topography, but if the topography is biased by 10 m (or 20 m), will it still be helpful? This is important to consider for flatter sites where the height uncertainty will be high due to the smaller convergence angles of the Planet images.
- When using the optimized, co-incident DEM for orthorectification with the bundle-adjusted L1B image pair, did this not result in removal or atleast some reduction of the *stereoscopic error*? Have the authors evaluated this? Ideally, this should have helped.
- The polynomial fit correction will require a good chunk of non-moving, static terrain distributed throughout the scene, which is important for users to consider on where this method is applied. Suppose a landslide is being studied in a glaciated area. In that case, this will likely be more difficult to apply as the glaciers will move, reducing the amount of static area that can be used for the presented sophisticated global ramp correction.

- How much of this polynomial fit step is required for L3B data from very similar perspectives?
- We should discuss how we are hampered by using just L3B data from common perspectives. What are we gaining from orthorectifying the L1B data from different perspectives using contemporaneous DEMs (i.e., how many new observations are added).

**Line by Line comments**

- The abstract is written very well, great work!
- Page 2 Line 43-44: This is a bit contradictory to your next sentence where you say that you then "carefully" select pairs from the same view directions, so how do you mitigate topographic errors arising from different view directions? Maybe clearly using bullets to describe the objectives of the study somewhere early on in the manuscript will help? You are doing a lot of cool stuff, and that should get the main space, which is getting lost in the current presentation. Something like:
  - Evaluate the different type of geolocation errors in different versions of **PS-SD** data (L1B, L3B, Basemaps)
  - Facilitate the use of images acquired from different perspectives in ground displacement tracking over dynamic terrain using an updated DEM derived from co-incident Planet imagery
  - Propose a workflow to carefully select L3B data for accurate ground displacement mapping?
  - Propose corrections on final displacement maps using polynomial fits to further reduce geolocation errors?
- Line 60 to 65: This could again be shortened, as the information can be presented better in the hopefully condensed study area section.
- Line 137-140: Maybe the line describing what RPC are and that they are used by Cubesat constellations can be skipped? RPCs are in widespread use now and are used by almost all satellite vendors who provide unrectified data.
- Rename Section 3.2: Again given the focus of the paper on PS2-SD data, do we need this section to be so lengthy? An alternative could be to describe in a sentence each all previous studies with old data (or maybe make a table of that with the sensors considered, the correction type, the number of images and science applications) and then let the readers figure out. The authors can then mention that none of these approaches have been able to correct errors in PS2-SD sensor, which is the main focus of the study.
- Line 234: Instead of going into all of this detail on how the data can be delivered, this could be simplified by saying we use green band due to xyz reasons, which corresponds to band x in PS2-SD data. We are not using products from older sensors, so why mention this granular detail about them?
- Line 477: What is meant by corrected L3B data? Has the polynomial fit been applied to the L3B data here?

- Section 5.3: In this section, I am a bit confused on what pairs where used to conduct this analysis. In Figure 14, was the L3B and L1B data selected for a pair acquired from a different perspective, or the similar perspective?
- Section 6.2.1 is important, thanks for conducting the analysis and sharing your findings!
- Line 530 to 533: could you show through a figure by what is meant by the misalignment in single vs multiple subframes? This is not clear to me in the current form.
- Line 537: What is the marginal lower accuracy (e.g., 1 m, 2 m, 3 m?) which the users should be comfortable with?
- Line 587 to 589: Thanks for conducting this analysis and sharing your results!

Sincerely,
Shashank Bhushan

---

## Author Comment (AC2)

Reviewer comments

Author comments

**Major comments**

1) The reading flow is not easy to follow, due to (a) long descriptions of methods, which could be greatly shortened (in addition, a general scheme at the beginning of section 4 would certainly be very useful to explain your processing chain from L3B or L1B images), (b) part of the description of the results included in the figure legends, (c) (too) many descriptive figures (I think a better selection of figures should be made. For example, figures 5 do not add much to understanding and can be placed in the supplementary material. Figures 4 and 6 illustrate the same effect), (d) the method section also includes results.

(a) We have shortened the method description and restructured it according to the suggestions of Reviewer 2, moving the description of the orthorectification error into a separate section and the analysis of stable pairs into the supplement. However, we would like to emphasize that a clear description of the applied method allows to more easily reproduce the processing steps that were taken.

(b) We do believe it is important that figure captions re-iterate and emphasize key findings and results. This will greatly enhance the context and readability of figures. Acknowledging your feedback, we have shortened figure captions in the result section including the previous Figure 12 (now Figure 10) and Figure 15 (now Figure 14).

(c) Previous Figures 5 and 7 have been moved to the supplementary material. We kept Figures 4 and 6 (now 4 and 5). Yes, they illustrate the same effect, but one figure is conceptional, the other shows real data and we feel it is important to demonstrate this to the reader. In accordance with Reviewer 2, we have added a sketch of the acquisition geometry to Figure 5 that allows to better link the figures.

(d) We have moved our analysis of the stable pair across the Siguas site to the supplementary material, and we agree that these are results. These show an experimental analysis to quantify the orthorectification error and do not match well with the presented results. This has further shortened the method section, and the section is now easier to follow.

2) The validation of the results is not really quantitative. Statistics should at least be provided to show the improvement of the different steps, for all sets of images. Here, a single histogram is shown for a correlation between 2 images (Figure 14). Why not extract a statistic (SD for example) from this histogram and compare it for the different steps for all pairs?

In addition, you could provide more quantitative validation by comparing your results with field measurements, which exist at least on the Siguas landslide (Lacroix et al., 2019).

To add a more quantitative assessment of our analysis, we have replaced the histograms shown in the initial manuscript by a scatter plot that shows the IQR across stable terrain where the landslide area was masked. This plot shows the offset before and after the application of the polynomial fit for all correlation pairs (n= 88 at the Siguas landslide and n=106 for Del Medio). We kept the histogram representation of the disparities before and after the correction, because it shows the shift of the entire distribution, but added them as a third column to the previous Figures 11 (now 9) and S5.

We have looked at the field measurements at the Siguas landslide, but we do not think that they complement our manuscript in a meaningful way because:

(a) the data presented by Lacroix et al., 2019 were acquired between November 2015 and May 2017. Earliest PlanetScope data is available from late 2016, so there is a temporal overlap of merely a few months.

(b) the current manuscript focuses on data acquired by the newer PlanetScope PSB.SD instruments which were not in orbit at the time the field measurements were taken. To compare velocity estimates to the GNSS data, we would have to work with data acquired by the older Dove-C (PS2) instruments, which were decommissioned in April 2022. Even though these data also suffer from orthorectification errors, a validation of will not be fully transferable to the newer sensors.

(c) In extending our quantitative analysis to include all pairs, and basing our assessment on the displacement over stable terrain, we adopt a validation approach widely recognized and utilized in other studies, such as Lacroix et al. 2023 and thus consider this as a robust alternative to field data comparisons.

3) The authors use 2 different processing approaches to extract displacement fields, using L3B or L1B images. I think there is a lack of clear discussion on which of these 2 approaches is more efficient. This discussion should be based on a more quantitative assessment of the errors on each of the processed pairs (see my previous comment). For me, this discussion should also include a systematic analysis (for all pairs) of subframe misalignment errors. The authors claim that they are reduced in the latest acquisitions. Could the author clarify why they have come to this conclusion, and when this improvement was made?

We have added a new section to the discussion titled "Comparison of L1B and L3B approaches", evaluating the different methods. We have not included a systematic analysis of sub-frame misalignment. Scenes with severe striping were excluded from the analyses presented in our manuscript. We have looked at generating robust methods to differentiate subtle and largely unpredictable striping patterns with smooth transitions from other co-registration errors in a quantitative way, but this is difficult due to the wide range of effects and errors. We cannot pinpoint a date when the improvement of the subframe misalignment was made, however, we observe that severe striping, as visible in Figure 3 D or Supplementary Figure S1, is most common among acquisitions made by the earliest PSB.SD Doves in the beginning of 2020.

**Detailed comments**

L15: "geoscientific": I would rather say "geomorphic".

Changed.

L36-37: "landslides are prone to orthorectification errors": It would be useful to quantify this orthorectification error. I suggest reviewing all the uncertainties associated with the use of PlanetScope data for landslide studies (Bradley et al. 2019; Mazzanti et al. 2020; Dille et al., 2021; Amici et al., 2022; Lacroix et al., 2023, ...).

We acknowledge the importance of quantifying orthorectification errors in landslide studies. Our detailed justification of why landslides are susceptible to orthorectification errors is presented in Section 3.1.1. The studies that are listed by the reviewer mostly do not consider the influence of orthorectification errors on their uncertainties, so we do not think a review of the presented uncertainties is very suitable to support that statement. We are aware that landslides with minor elevation changes compared to Siguas and Del Medio landslides will show a much lower impact from orthorectification errors. In the revised manuscript, we have therefore specified that landslides orthorectification errors are common for landslides with relief that have significantly altered the landscape over time:

Before: … landslides are prone to orthorectification errors …

Now: … landslides that have significantly altered the landscape over time are prone to orthorectification errors …

L75: It would be interesting if you also mentioned that monitoring already exists on the Siguas landslides, which could be used to validate your results (Lacroix et al., 2019). See my main comment no. 1.

See our reply to main comment no. 2.

2.1: Are there independent estimates of the speed of the Del Medio landslide?

No, unfortunately not, but we are in the process of setting up GNSS stations in the area.

Figure 1: As things stand, the black and blue lines mentioned in the legend are difficult to see. Is there any real point in showing the road network?

We have changed the catchment divide between the Central Andes and the foreland region to white. We keep the road network in the figure, because it shows important aspects of the infrastructural network and allows to identify the geographic location of the landslides.

L110: "NIR measurements are stored at the green pixels of the RGB Bayer-mask (Planet, 2022a). "This is not clear. Besides, is there any point in knowing this information? In general, I think authors should simplify their text to make it easier to read (see my main comment No. 1).

We have removed that sentence.

L114: "NIR band is captured at a different time": Can you specify what the timeframe is?

When the consecutive frame is captured, so approximately 1-2 seconds. We have replaced "different time" with "a few seconds later" to make that clear.

Figure 3: This is a nice figure to show the different errors. I would simply reverse the order of the legend so that it corresponds to the order of the sub-figures: (1) DEM error (A), (2) striping errors (B, C, D), (3) overall shifts between scenes (C, A), (4) stereoscopic errors (D).

We re-organised Figure 3 to fit the order in which the different errors are presented in the text: (1) orthorectification error, (2) stereoscopic effects, (3) global shift and ramp errors, (4) stripes. We also picked new examples that highlight each error individually.

L155-156: It should be noted that the error associated with a global offset is classically corrected for slow slide studies using PlanetScope, which significantly reduces the errors (see also my comment on the uncertainty associated with PlanetScope images of slow slides l36-37).

We added that in the section on previous approaches improving the co-registration accuracy:

Lines 191-194: The proposed mitigation strategies include registering PlanetScope scenes to high-resolution reference imagery (Dille et al., 2021), subtracting the median displacement estimated over stable terrain (Lacroix et al., 2023), both of which efficiently remove global shifts, and the fitting of polynomials (Kääb et al., 2017, 2019; Feng et al., 2019).

L240: L1B images are also available in clipped format.

To the best of our knowledge, they are not. We have been in contact with the Planet Support particularly about this issue and they pointed us to this article: Why isn't the clip tool available for a basic scene in Planet Explorer?

Figure 5: «Scenes acquired from an opposite view direction at high view angles are strongest affected by orthorectification errors.» Opposite view direction to what? why should orthorectification errors be stronger with some specific viewing angles? This sentence is not clear.

Opposite of each other, i.e. a left- and right-looking satellite at 5° off-nadir. However, for clarity reasons, we have removed this sentence from the figure description to focus on the acquisition parameters only.

Furthermore, Figure 5 may not be necessary to understand the study. In fact, it is mentioned only once in the text. Could you place this figure in the supplements?

We placed Figure 5 in the Supplementary Material.

Lines 250-287: This section can really be reduced. Figure 4 illustrates this well. I also wonder if this section should not be mixed with section 3.1, when the effect of orthorectification errors is illustrated in figure 2. In this section, you do not propose a method for reducing this error, but you do illustrate it. In my opinion, it should not be included in the "data and methods section".

We greatly streamlined the description of the orthorectification error and included it in section 3.1 as suggested.

Reorganisation: Lines 288 to 313: I get the impression that this section is a bit vague and that the flow is not easy to follow because it's a mixture of methods and results. If I understand you correctly, you identify the acquisition parameter that allows you to form pairs and reduce uncertainties while correlating them. I have the impression that you could state this much more clearly and separate the methods from the application to the data. The choice of figures also makes things less easy to follow: Figure 6 is closely related to Figure 4 in terms of illustrating the problem (perhaps one of the two figures could be placed in the supplementary material?) Figure 7 is an application of the methods that shows the important effect of the actual azimuth of observation. Figure 8 shows your results once the groups have been created.

This section has been greatly restructured. We have moved our analysis of the relationship between orthorectification error and true view angle difference on the basis of short temporal baseline pairs across the Siguas landslide to the Supplementary Material along with former Figure 7. The description of the orthorectification errors was relocated from the method section to section 3.1.1 describing the spatial patterns of orthorectification errors in PlanetScope data. We would like to show both Figures 4 and 6 (now 4 and 5), as we think the conceptual sketch is instructive, but it is also necessary to see the effect in actual data.

In the same section, it is not clear, once your pairs are created, how you will use them to create a time series of movements. In fact, there is no possible relationship between the different groups of images. How do you put them back together? I have the impression that a general diagram at the beginning of section 4 would be useful to explain your processing chain based on images L3B or L1B.

While no correlation pairs are formed between images from different groups, the results are still spatially related. Given the ~2-pixel geolocation accuracy, measurements from different disparity maps may exhibit an offset of approximately 6 m. However, for landslides spanning several hundred meters in diameter, neighboring pixels are likely to exhibit similar movement patterns. The key is ensuring the offset estimation for a given pixel is reliable which is achieved by reducing orthorectification errors and other factors affecting co-registration precision.

We have significantly restructured our method section for enhanced clarity and therefore do not see the need for a general diagram, especially considering the feedback about too many conceptual figures. If in doubt, our workflow is clearly documented in our accompanying GitHub repository.

Line 375: I have the impression that the method you describe has already been described and used by Berthier et al (2007). You can certainly simplify your text by refering to it.

We read the Berthier et al. 2007 paper and their approach is to minimize the standard deviation of the difference between the two DEMs outside glaciated regions, while we minimize the sum of displacement between pixel correspondences when projected from one image to the other using the PlanetScope DEM at a given position. We have, however, further simplified the text in order to shorten this section.

Line 394: MPIC-OPT is not strictly a correlator but a processing chain that does more than correlate. The correlator behind MPIC-OPT is Mic-Mac (Rupnik et al., 2017).

Thank you for clarifying that. We have included the reference to Mic-Mac in the revised manuscript.

Line 399: Why do you use 35x35 pixel windows? Did you do several trials before choosing?

Yes, we have experimented with several window sizes and matching algorithms. Results within a ± 10 pixel range are comparable at the study sites. Smaller windows generally produce noisier disparity maps, but can better capture large magnitude displacements and spatial variability. As both the Siguas and the Del Medio landslide are slow-moving targets, we preferred to use a larger correlation kernel to reduce noise and obtain smoother displacement estimates.

Line 399: I would also delete "slightly larger correlation".

We removed that.

Figure 11: There is no reference to sub-graphs C and D.

Thank you for noting that. We now reference the entire figure in line 335.

Line 417: You mention striping effect due to the misalignment of the subplots, but I assume that your polynomial (line 420) does not correct this effect. How effective is your polynomia at correcting other artefacts when you have such effects?

That is true – stripes are not corrected for by the polynomial fit. We have restructured our method section to make clear which approach is used to correct for what pattern of co-registration error. Scenes that show severe striping in the derived disparity maps were sorted out (see section 4.5). If slight striping is still present, the polynomial fit still is very efficient in eliminating other effects, as visible in former Figure 11 (now Figure 9) in the dy component.

Line 435: The speed of the landslide may vary over time, but you are assuming here that the speed is constant over the period in question. This needs to be made clear. In fact, you mention this transient in the caption to Figure 12. This assumption could be verified by comparing satellite measurements with field data. These validation data are available on the Siguas landslide (Lacroix et al., 2019). See also my following comment on the results validation.

Yes, we address this in the discussion (section 6.1), but have clarified that the offset is averaged over the time that lies between the acquisition of first and secondary image:

Lines 441-442: It is important to note that the offset estimates obtained from correlating two images captured at different points in time represent an average offset over that time span. The velocity is therefore assumed to be constant over the considered period. The actual displacement may occur more abruptly …

By the way, your Figure 13 seems to show that the standard deviation of velocity is significantly reduced when using manually orthorectified 1B products compared to correlating 3B products from the same group. To me, this means that the higher standard deviation observed on the landslide in Figure 12 does not come from transient motion but rather from orthorectification errors in the 3B products, even if you select the pairs. Can you comment on this point?

Yes, the strong reduction of standard deviation shown in Former Figure 13 (now 11) is related to the reduction of orthorectification errors as the L1B images are mapprojected using an updated reference surface. This is our motivation for the proposed correction approach. The transient offset is only responsible for the slightly elevated standard deviation at the lower part of the Siguas landslide in groups 1-3, see lines 379 to 382:

"Slightly higher standard deviations for groups 1-3 at the landslide toe are likely related to transient changes in velocity. In contrast, when PlanetScope scenes are selected randomly (group 4), variations in view angle and satellite azimuth, combined with outdated DEM surfaces used for orthorectification, result in misprojections over the landslide surface."

Line 436: remove an "only".

Done.

Figure 15: I don't quite understand how you obtain these time series. I understand that you correlate either the L1B data from the same group or the L3B data from group 4, but do you correlate them all within each group or do you only correlate those that are separated by the shortest time to see the transients?

We correlate only within a group and here we correlate all pairs that have a minimum time difference of 180 days to ensure that enough displacement has accumulated to reach the detection limit. Consequently, the data points indicate the average displacement across variable time scales. To get individual time steps, an inversion approach such as SBAS could have been applied. We are currently working on a separate manuscript that evaluates the inversion approaches for optical data. This is a different topic and beyond the scope of this work.

Line 541 : The correct reference is Lacroix et al., 2019 not 2015

Corrected.

Line 550-553 : Are you removing the low quality pixels from the « good pixel map » ? In this case it should highly remove the changes in soil occupation, and therefore the errors. Are you not sure that the higher standard deviation with time in the Siguas case study is not caused by the motion of the landslide that occupies a quite important area of the image ?

Yes, we always remove the low quality matches using the "good pixel mask". This does remove a large portion of the high-offset pixels within the agricultural areas towards the NW of the study area, but not all of them. To be absolutely sure that the higher standard deviations are not related to the landslide, we reran the analysis and masked the landslide, so that statistics were only calculated across stable terrain.

Rather than a hypothetic section on the «Transferability to other regions and targets», I would have rather see a discussion on which of the L1B or L3B processing should we use (See my major comment n°3). From the Figures you show, manually orthorectified L1B sounds more efficient, except for the sub-frames alignment. However it lacks this analysis for all the scenes processed. Furthermore, is the sub-frame alignment really better now ?

We would like to keep the section on transferability to other regions and targets in the main manuscript as we believe it is important to readers. Nevertheless, we have added an additional section, titled "Comparison of L1B and L3B approaches" (see above), where we discuss advantages and drawbacks of both approaches.

Figure 15 and lines 610-612 : Can you explain how you obtain the shown uncertainties and add uncertainties on your velocity estimations ?

In the initial manuscript, the uncertainties presented in Figure 15 (now 14) were based on the standard deviation of all pixel values within the landslide area. However, given the frequent use of displacement across stable terrain as a quality metric, we have reevaluated these uncertainties. The data points still reflect the mean velocity inside the landslide area, while the uncertainties represent the mean velocity across stable terrain (areas outside the landslide mask). We have also updated Figure 14 in the regard that we no longer separate the Siguas landslide into head and toe regions. To emphasize the spatial variability of velocity across the landslide, we instead incorporated a map view in Figure 14 A, showcasing the average velocity derived from all correlation pairs within the landslide mask.

The updated Figure 15 (now 14) is shown below:

[Figure]

---

## Author Comment (AC3)

Reviewer comments

Author comments

**Major points**

● I believe early on in the manuscript, it should be mentioned that the analysis is focussing on PlanetScope SuperDove data. Then the authors can potentially consider reducing the information presented on the earlier generation of PS constellation (current section 3.), and condensing the information on previous efforts which have tried to work with older PS data (Section 3.2). Mentioning some of the previous approaches is valuable, but I do not think a detailed description of those is required. More comments related to this point are also provided in other comments below.

We now emphasize that our work focuses on the newer PlanetScope generation in the Introduction section (see Lines 44-45). The orthorectification error, however, also affects data acquired by the older PS instruments. We therefore provide information on all Dove generations, but we have greatly shortened the section on previous efforts (see our responses to the related comments below).

● Consider reducing the dense text on the background of the two landslides. I agree the event description is important, but describing the general characteristics of the events and then pointing the users to published papers for more in-depth details will work better here, as the current paper does not focus on the science of landslides per se.

We agree and have condensed this section.

● Section 3.1 is again pretty dense. One option could be to break into 2 separate sections. The first could be renamed as section 3.1: Expected relative geolocation accuracy, which succinctly describes the geolocation accuracy which is expected in Planet data, quoting numbers from the Planet documentation and previous studies. The second section could describe in more detail the spatial pattern of the typical relative geolocation offset, leading up with the figure 3, describing the four main types of errors. I think there are 4 main issues shown in figure 3 are:

  ○ Error due to dynamic/outdated topography in displacement maps obtained from image pairs acquired from different orbits

  ○ Error due to stereoscopic affects in the y-direction (again more for pairs collected from different orbits?)

  ○ A general global shift which can be corrected by removing the median x and y shifts over static surfaces (this could be present in image pairs collected by both the same and different orbits)

○ Striping effects in the y-direction due to sub-frame misalignment (will be present in both L1B and L3B data, and potentially for image pairs acquired from both similar and different orbits).

Based on the order you chose to introduce these errors, have small subsections or bullets for these 4 points in the new section. It will then be very clear to the reader that these are the four errors that the authors are going to tackle in the manuscript. Then, maintain this order when you propose corrections, describe results, and conduct discussions.

There should also be a clear distinction between which correction corrects for what error. So maybe we the corrections are introduced, their section headers could contain information about which of the four errors is being corrected?

Along the same lines, it would be useful to plot the image acquisition geometry skyplot in the third column for each of the two pairs in Figure 3 (as in figure 8), so that it helps us in bringing out the effect of acquisition geometry on some of these errors?

Thank you for these suggestions. We have followed this and now list and explain the 4 spatial patterns of error observed in the disparity maps (orthorectification error, stereoscopic effects, ramp errors and global shift, stripes) in section 3.1. We have also reorganized our method section accordingly and now have separate subsections that describe how to mitigate what error.

We really liked the suggestion of adding a skyplot and did so, not for Figure 4, because not all co-registration errors are related to large view-angle differences, but to Figure 6 (now 5) showing the orthorectification error for an image pair acquired from a different and common perspective (we believe the reviewer meant Figures 4 and 6 instead of Figures 4 and 8). Our new Figure 5 is shown below:

[Figure]

● Current section 4.2 is very long, and the order in which information is presented can be improved. You already talked about how outdated DEM affects displacement mapping in the current section 3.1, ideally the your conceptual figure and text belong there, and not in the methods.

We agree that a joint description of concepts and methods is not ideal. We have integrated the description of the orthorectification error and conceptional figures in section 3.1 (see our reply to major point 3).

● Similarly, the authors talk about the discussion of using data from a .json file or the scene metadata.xml. All of this is important to consider, but it disrupts the scientific/methodological flow of the paper. Maybe some of this could be transferred from the manuscript to the github repository readme or something?

We have moved this information to the documentation in the accompanying github repository.

● DEM generation section have no mention of how accurate the output DEM is? I understand the authors want an update topography, but if the topography is biased by 10 m (or 20 m), will it still be helpful? This is important to consider for flatter sites where the height uncertainty will be high due to the smaller convergence angles of the Planet images.

We have carried out an additional analysis to address this question. We imposed an artificial (known) elevation bias onto a DEM and compared the effect on disparity maps derived from an orthoprojected L1B pair acquired from different perspectives. We find that a vertical DEM bias of 1 m results in an orthorectification error of on average 0.056 pixel or 0.168 m in the cross-track (dx) component, while the along-track disparities (dy) appear unaffected. If a 20 m bias in the topography is still useful depends on the analyzed scenario. If you can tolerate ~3.36 m uncertainty in velocity estimates because the magnitude of displacement largely exceeds that value, it will not be a problem. However, measurements of slow slides with annual velocities of only a few meters will suffer much more from a larger DEM bias.

We have described this analysis and findings in the newly added sections 4.7 (Assessment of orthorectification error magnitude in response to DEM changes) and 5.3 (Magnitude of orthorectification errors).

● When using the optimized, co-incident DEM for orthorectification with the bundle-adjusted L1B image pair, did this not result in removal or at least some reduction of the stereoscopic error? Have the authors evaluated this? Ideally, this should have helped.

Yes, we have tested bundle adjustment but observed no enhancement in the quality of the disparity maps. In many cases, it degraded the co-registration accuracy even further.

Here is an example of an L1B image pair (2021-05-16 and 2023-05-11) mapprojected with and without bundle-adjust-prefix. The larger topographic effects after the bundle adjustment are visible by the larger color gradient.

[Figure]

Consequently, we decided to use the original RPCs from Planet without any bundle adjustment to carry out the map projection. This decision aligns with a discussion in the Ames Google Group, where bundle adjustment also did not resolve systematic errors: https://groups.google.com/g/ames-stereo-pipeline-support/c/MTVVV00Qf0I/m/fXTig3rzAgAJ

● The polynomial fit correction will require a good chunk of non-moving, static terrain distributed throughout the scene, which is important for users to consider on where this

method is applied. Suppose a landslide is being studied in a glaciated area. In that case, this will likely be more difficult to apply as the glaciers will move, reducing the amount of static area that can be used for the presented sophisticated global ramp correction.

That is true and we have noted that in the discussion lines 482-483 . We have added that glaciated areas pose a particular challenge.

Lines 483-484: Similarly, terrains with widespread movement, e.g. glaciated zones, present challenges for ramp correction.

● How much of this polynomial fit step is required for L3B data from very similar perspectives?

We do not find a clear link between the difference in satellite perspective and the need for fitting a polynomial. For example, the image pair displaying a ramp error in updated Figure 3 C only has a view angle difference of 0.4°. Instead, we assume that this issue is more related to how well the camera position was constrained and how many tier points were found, which can vary from scene to scene.

● We should discuss how we are hampered by using just L3B data from common perspectives. What are we gaining from orthorectifying the L1B data from different perspectives using contemporaneous DEMs (i.e., how many new observations are added).

We have added a new section to the discussion about the advantages and limitations of the L1B and L3B approaches (section 6.3) and a new supplementary Figure (S13) showing the availability of suitable PlanetScope scenes (cloud-free, full AOI coverage) and their true view angle.

**Line by Line comments**

● The abstract is written very well, great work!

Thank you!

● Page 2 Line 43-44: This is a bit contradictory to your next sentence where you say that you then "carefully" select pairs from the same view directions, so how do you mitigate topographic errors arising from different view directions? Maybe clearly using bullets to describe the objectives of the study somewhere early on in the manuscript will help? You are doing a lot of cool stuff, and that should get the main space, which is getting lost in the current presentation. Something like:

○ Evaluate the different type of geolocation errors in different versions of PS-SD data (L1B, L3B, Basemaps)

○ Facilitate the use of images acquired from different perspectives in ground displacement tracking over dynamic terrain using an updated DEM derived from co-incident Planet imagery

○ Propose a workflow to carefully select L3B data for accurate ground displacement mapping?

○ Propose corrections on final displacement maps using polynomial fits to further reduce geolocation errors?

Thanks for pointing this out. We have followed your advice and have replaced the former description of our contributions with the following bullet points to present them more clearly:

1. Examine the different sources of errors compromising co-registration accuracy between PlanetScope scenes, particularly those captured by the latest PSB.SD instruments.
2. Present a workflow to mitigate the orthorectification error through a careful selection of correlation pairs based on common satellite perspective (jointly determined by the satellite's look direction, view angle, and motion direction) for orthorectified L3B data.
3. Enable the use of images acquired from different perspectives through manual orthorectification of unrectified L1B data based on an updated DEM derived from co-incident Planet imagery.
4. Propose corrections of the displacement maps through fitting polynomials to further reduce co-registration errors.

● Line 60 to 65: This could again be shortened, as the information can be presented better in the hopefully condensed study area section.

We have removed the last two sentences of the introduction and included this information in the description of the test sites.

● Line 137-140: Maybe the line describing what RPC are and that they are used by Cubesat constellations can be skipped? RPCs are in widespread use now and are used by almost all satellite vendors who provide unrectified data.

We have removed the line about the RPC use by Cubesat constellation and RPC bias compensation to shorten the paragraph but would like to retain the short description of RPCs for readers who are not familiar with the concept.

● Rename Section 3.2: Again given the focus of the paper on PS2-SD data, do we need this section to be so lengthy? An alternative could be to describe in a sentence each all previous studies with old data (or maybe make a table of that with the sensors considered, the correction type, the number of images and science applications) and then let the readers figure it out. The authors can then mention that none of these approaches have been able to correct errors in PS2-SD sensor, which is the main focus of the study.

We have greatly shortened Section 3.2, summarized the previous approaches and stressed that in our work, we focus on the newer PSB.SD instruments.

● Line 234: Instead of going into all of this detail on how the data can be delivered, this could be simplified by saying we use green band due to xyz reasons, which corresponds to band x in PS2-SD data. We are not using products from older sensors, so why mention this granular detail about them?

True. We have modified the line accordingly.

● Line 477: What is meant by corrected L3B data? Has the polynomial fit been applied to the L3B data here?

Yes, with corrected L3B we refer to the processing steps where the polynomial fit has been applied. In response to the feedback from Reviewer 1, we have completely rewritten this section and made sure to explicitly state which data at which processing state we refer to.

● Section 5.3: In this section, I am a bit confused on what pairs were used to conduct this analysis. In Figure 14, was the L3B and L1B data selected for a pair acquired from a different perspective, or the similar perspective?

Previous Figure 14 showed histograms estimated from a pair taken from different perspectives. However, in  response to the feedback from Reviewer 1, we have replaced that Figure by a scatter plot showing the IQR across stable terrain for all image pairs before and after the application of the polynomial fit.

● Section 6.2.1 is important, thanks for conducting the analysis and sharing your findings!

We are glad to hear that you found this analysis useful.

● Line 530 to 533: could you show through a figure by what is meant by the misalignment in single vs multiple subframes? This is not clear to me in the current form.

An example of what the misalignment of single vs. multiple subframes looks like is given in Figure 3B and 3C in the original manuscript. Misalignment of multiple subframes results in a regular striping pattern (Figure 3B). In contrast, in Figure 3C we observe that only the lowermost part of the disparity map is offset. The latter scenario can be approximated by a second or third order polynomial, but multiple stripes cannot. We agree, however, that this statement may be confusing and because it is a rather rare case, we have removed the sentence about the compensation of single frame misalignment from the manuscript and also chose different examples for Figure 3.

● Line 537: What is the marginal lower accuracy (e.g., 1 m, 2 m, 3 m?) which the users should be comfortable with?

That depends on how many sub-frames were misaligned in an image and how well the transitions were smoothed. From our experience, we see that the magnitude of the striping effect is typically at or below 1 pixel (3 m). Considering the spread of disparities estimated across stable terrain as a quality indicator, Figures 13 and 15 in the revised manuscript show that the IQR of the dy component is only about 0.25 pixel (0.75 m) larger than that of the dx component. On this basis, we define the "marginal lower accuracy" as < 1 pixel in the revised manuscript.

● Line 587 to 589: Thanks for conducting this analysis and sharing your results!

Thank you!

---

## Author Response (AR3)

Dear Giulia Sofia,

We have taken into account all comments raised by the reviewer Pascal Lacroix and have modified our manuscript accordingly.

Please find our replies to all individual comments attached.

Best regards,

Ariane Mueting

on behalf of the authors

Reviewer comments

Author comments

Authors made substantial changes to improve the quantitative assessment of their processing, and I would like to thank them for that effort. I also feel the flow of the reading improved thanks to the modifications realized. I feel the manuscript is almost ready for publication and requires two moderate modifications and then only minor changes listed below.

Thank you for taking the time again to review our manuscript. Your feedback is very much appreciated.

Moderate comments :
Introduction :
As the introduction is turned, it gives the impression that there is no previous work to reduce orthorectification errors in PlanetScope images. In fact, the authors chose to describe this work in part 3.2 'Past Approaches to improve scene-to-scene coregistration' and in part 4.4.2. I understand that it is first necessary to describe the geometry of PlanetScope acquisitions before describing the approaches used. Having said that, I think it would be more honest to put somes sentences about previous works in the introduction, for example just after line 41, to show that your study fits into a general context. The whole section 3.2 would fit perfectly in the introduction. If you don't think so, please summarize it in a few sentences to add it in the introduction.

We have moved the entire section 3.2 to the introduction, as suggested.

Discussions of the velocity time-series on the two landslides investigated :
In the present state, authors decided to show the time-series of velocity, without much interpretations. The interpretations of the time-series are only expressed in the conclusions in two sentences. I still believe (as my previous round of comments mentioned) a more in-depth discussion of the observed velocities is required. Indeed, the authors decided to submit their study in E-surf, which is a journal dedicated to better understand surface processes. So the application seems to me quite important.

We have elaborated more on the interpretation of the observed velocities, in particular on the potential seasonal control of the Del Medio landslide, see lines 466-474 in the revised manuscript. The focus of our work, however, is to assess the impact and mitigation of errors in satellite-based measurement of surface displacement. This is the reason why we keep the interpretation of the specific velocities at our test sites rather short.

Minor comments :

Abstract : Please change the range of velocity of Slow-moving landslides for 1-100 m/yr to be consistent with the velocity of the landslides you are investigating (Siguas reaches 40 m/yr). This also appears several times in the manuscript (Line 155, elsewhere...).

We changed all instances to 1-40 m/yr.

Lines 164-165 : change « make it nearly impossible…. » for « lead to large uncertainties on the horizontal displacement ».

Changed.

3.1.1 : if you choose to still use both EW and dx (NS and dy) please define them first time you use them (In its present state it is described in Caption of Figure 6). I still think it would be easier to have only EW/NS and remove dx/dy.

We have now indicated that we refer to EW displacement as dx and NS as dy upon first occurrence (see lines 244-245 in the revised manuscript). Both terms are known in the scientific community so we choose to keep them.

3.2 : This section is an introductory paragraph to explain the previous works realized on the investigated topic. I believe this should be placed in the introduction.

We have moved this section to the introduction, see our response to moderate comment #1.

Lines 281 : Could you precise the dates of acquisitions of the 2 DEMs (Copernicus and NASADEM) ?

We have added the acquisition years for both DEMs.

Lines 287-296 : These sentences could be added in the introduction to explain the previous approaches.

We have left these sentences in place, because we believe other studies using PlanetScope data for DEM generation are best explained jointly with our own 3D processing workflow. However, we have moved the entire section 3.2 to the introduction to provide a reference to previous approaches earlier in the manuscript, as suggested by the reviewer (see our response to moderate comment #1).

Lines 329-330 : The sentence is not clear. Either reformulate or explain the wording « image space » and « object space » .

We have added explanations (see lines 323-326 in the revised manuscript):
"When two unprojected L1B scenes are correlated in image space (pixel coordinates, i.e. rows and columns) and the obtained correspondences are transferred into object space (geographic coordinates, i.e. longitude and latitude) using the given RPCs and a DEM, the displacement between the projected pixels should be zero."

Figure 9 : Could you please change the order of the image subsets B/C E/F to be consistent with the order of the Figure 8, where results with the NASADEM appears before the results for Copernicus ?

We have reordered the panels of Figure 9 accordingly.

Line 345 : « disparities » is still appearing. Can you change that ?

All occurrences have been changed.

Line 405 : « displacement velocity » : not clear. Do you mean « velocity magnitude » ?
Line 408 : replace « velocities » for « magnitudes »

We have changed all occurrences of "displacement velocity" to "velocity magnitude". Velocity is a vector with magnitude and direction.

Figures 11 and 12 : « displacement velocity » : not clear. Do you mean « velocity magnitude » ?

Yes. We have changed all occurrences of "displacement velocity" to "velocity magnitude", see previous comment.

Figure 15D : Could you zoom the y-axis to better see the kinematic variations of the landslide ? For instance between 0 and 10 m/yr ?

We prefer to keep common y-axis limits for both landslides to allow for a better visual comparison of velocity magnitudes and error bars.

Conclusions : the interpretation of the time-series are only expressed here in two short sentences. I still believe a more in-depth discussion of the observed velocities are required, as the authors decided to show the time-series of velocity and submit their study in the E-surf journal, dedicated to surface processes.

See our response to moderate comment #2.